# Geodesic Calculus on Implicitly Defined Latent Manifolds

Florine Hartwig [1]   Josua Sassen [2]   Juliane Braunsmann [3]   Martin Rumpf [1]   Benedikt Wirth [3]

## Abstract

Latent manifolds of autoencoders provide low-dimensional representations of data, which can be studied from a geometric perspective. We propose to describe these latent manifolds as implicit submanifolds of some ambient latent space. Based on this, we develop tools for a discrete Riemannian calculus approximating classical geometric operators. These tools are robust against inaccuracies of the implicit representation often occurring in practical examples. To obtain a suitable implicit representation, we propose to learn an approximate projection onto the latent manifold by minimizing a denoising objective. This approach is independent of the underlying autoencoder and supports the use of different Riemannian geometries on the latent manifolds. The framework in particular enables the computation of geodesic paths connecting given end points and shooting geodesics via the Riemannian exponential maps on latent manifolds. We evaluate our approach on various autoencoders trained on synthetic and real data.

## 1. Introduction

In machine learning, extracting low-dimensional data representations is a classical problem, motivated by the manifold hypothesis that many high-dimensional datasets, such as images, lie on or near low-dimensional submanifolds. Approaches range from classical manifold learning methods, such as Isomap (Tenenbaum et al., 2000) and Diffusion Maps (Coifman et al., 2005), to neural network-based methods, including autoencoders, their probabilistic variants (e.g., variational autoencoders, VAE) (Kingma & Welling,

2013), and Generative Adversarial Networks (Goodfellow et al., 2014).

These ideas remain central in modern machine learning, for instance, in word embeddings for large language models (Devlin et al., 2019) or autoencoders in diffusion models (Rombach et al., 2022). Low-dimensional representations of the data manifold are crucial for high-performing generative models, yet their rich information (e.g., intrinsic dimension, topology, or point proximity) is rarely used explicitly. Instead, they are primarily considered an intermediate compression step.

In contrast, shape analysis extensively exploits manifold representations of geometric data. Riemannian manifolds—manifolds with a local measure of length—are a standard tool for modeling collections of shapes called shape spaces. Derived geometric operators are central for celebrated methods like LDDMM (Beg et al., 2005), enabling applied tasks to be phrased in terms of Riemannian calculus, e.g., shape interpolation via computing interpolating geodesics and shape extrapolation via the exponential map.

Yet, evaluating Riemannian operations on shape spaces is often computationally expensive, partly due to high dimensionality. Autoencoders could efficiently parametrize low-dimensional shape submanifolds, but their latent spaces typically lack explicit geometric structure. This highlights an open challenge in manifold learning: equipping latent spaces with geometric structure and practically usable geometric operators.

Previous work has focused on learning underlying structures, e.g., manifold representations (Arvanitidis et al., 2018) or Riemannian metrics (Gruffaz & Sassen, 2025). The computation of geodesics on latent manifolds is often restricted to specific autoencoders (Arvanitidis et al., 2022) and specific choices of the metric. Practical computation of geodesic interpolation between given endpoints or geodesic extrapolation for some initial position and direction that can be seamlessly integrated in an existing autoencoder pipeline remains challenging. There are two essential properties to be achieved. First, the computed paths must adhere to the latent sample distribution. Second, for non-isometric embeddings a problem-dependent metric must be additionally incorporated. To address this, we make the latent manifold's geometry accessible via an implicit representation based

---

[1] Institute for Numerical Simulation, University of Bonn, Germany [2] Centre Borelli, ENS Paris-Saclay, France [3] Institute for Computational and Applied Mathematics, University of Münster, Germany. Correspondence to: Florine Hartwig <florine.hartwig@uni-bonn.de>.

*Proceedings of the $43^{rd}$ International Conference on Machine Learning*, Seoul, South Korea. PMLR 306, 2026. Copyright 2026 by the author(s).

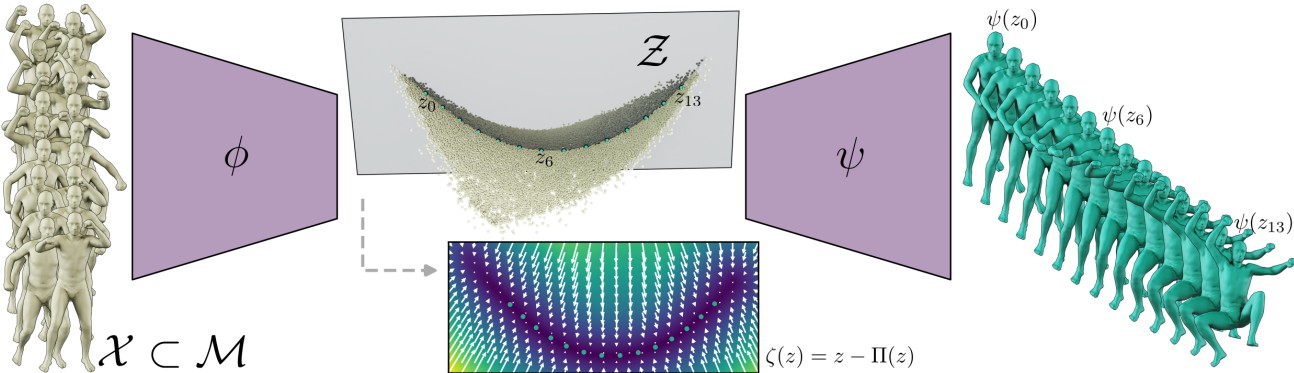

*Figure 1.* Training an autoencoder $(\phi, \psi)$ with data $\mathcal{X}$ lying on a manifold $\mathcal{M}$ yields a low-dimensional latent manifold $\mathcal{Z}$. Using a denoising objective, we learn an implicit representation $\zeta$ of this manifold (color-coding on the 2D slice from blue to yellow indicates $|\zeta|$) based on a projection $\Pi$ onto $\mathcal{Z}$ (white arrows, rescaled). Furthermore, we introduce a practical geodesic calculus on this representation, enabling, e.g., shape interpolation using latent manifolds (green dots and shapes).

on a learned projection that minimizes a denoising objective. Furthermore, we introduce a time-discrete variational geodesic calculus which allows for different metrics and is suitable for imperfect implicit representations. This tool can be used in diverse applications and is already known to approximate ground truth geodesics well in the case of perfect implicit representations. Figure 1 shows an illustrative example for both desired properties. In addition the toy example in Figure 2 motivates the need of an actual representation of the latent manifold geometry for geodesic interpolation purposes. Building on a suitable time discretization which proved effective for Riemannian shape spaces (Rumpf & Wirth, 2015), we hope to open up new ways of using latent representations in machine learning in general and enable new reduced-order methods for shape spaces in particular.

**Contributions.** In summary, we make the following contributions:

- We introduce a time-discrete geodesic calculus for imperfect representations of implicit latent manifolds and provide a framework to compute realistic interpolation and extrapolation on these manifolds. For regular, smooth data manifolds this framework corresponds to geodesic inter- and extrapolation with respect to a Riemannian metric.

- We suggest minimizing a denoising objective to learn an approximate projection on latent manifolds with unknown codimension.

- We evaluate our approach on various autoencoders trained on synthetic and on real data.

- We provide code in an easy-to-use fashion for different metrics and implicit representations https://github.com/flrneha/LatentGeodesics.

### 1.1. Related Work

**Latent Space Geometry.** Their widespread use has made the latent space of autoencoders a prime object of study, and equipping them with an appropriate notion of geometry is a major goal. Shao et al. (2018) compute interpolations and other geometric operations on the data manifold by parametrizing it via the decoder. However, they ignore that the actual latent manifold may have nonzero codimension in latent space. Chen et al. (2018) pursue a similar idea for VAEs, where they additionally modify the pulled-back metric to generate high cost away from the data manifold. This underlying idea was concurrently pursued by Arvanitidis et al. (2018) based on previous work by Tosi et al. (2014) on Gaussian process latent variable models. They further extended this idea to pulling back non-Euclidean metrics (Arvanitidis et al., 2021), to pulling back the Fisher–Rao metric on densities (Arvanitidis et al., 2022; Lobashev et al., 2025), and to using Finsler metrics on latent spaces (Pouplin et al., 2023). In contrast to these works, we propose to encode the latent manifold as an implicit submanifold and derive a discrete geodesic calculus from this description. This allows both to compute paths that follow the latent manifold, i.e. the data support, and also to incorporate application-dependent metrics. Sun et al. (2025) also learn an implicit representation of the latent manifold, however, they use it to modify the metric on the latent space and pull this modified metric back to the data space via the encoder. We will perform all optimizations directly on the latent manifold to maintain the advantages of its low dimensionality. In Appendix C.3 and Appendix C.4 we further discuss and compare with different methods that compute paths following the data density.

**Discrete Geodesic Calculus.** One of the most fundamental tasks in Riemannian geometry is the computation of geodesic paths, either solving the system of geodesic dif-

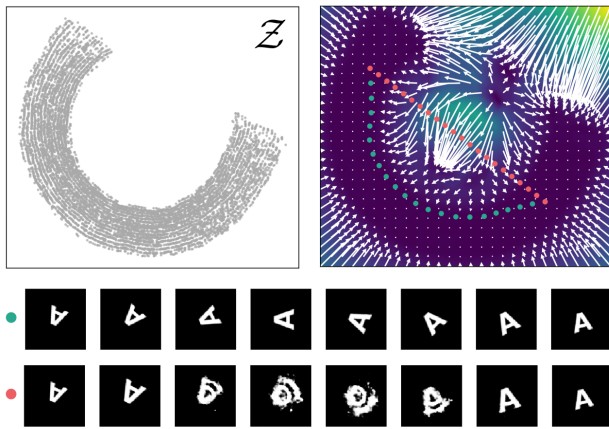

**Figure 2.** Standard autoencoder trained on images of a rotated and scaled letter 'A' with two-dimensional latent space. Comparison between geodesic interpolation (green) using our method and linear interpolation (red) showing below every third decoded image.

ferential equations via numerical integration techniques for given initial data, as in (Beg et al., 2005), or minimizing the so-called path energy over paths with prescribed endpoints. The latter variant is based on a suitable discretization of the path energy depending on the type of Riemannian space. In Section 3 we will follow this time discretization paradigm. By Rumpf & Wirth (2015) it was developed into a comprehensive time-discrete geodesic calculus on shape spaces (including a corresponding convergence analysis for vanishing time steps), and it was applied in different contexts such as curves (Bauer et al., 2017), discrete surfaces (Heeren et al., 2014), or images (Berkels et al., 2015). In Appendix C.1 and Appendix C.2 we more extensively compare different manifold representations and numerical discretizations.

**Neural Implicits.** Implicit representations of geometric objects using neural networks have emerged as a new paradigm in computer graphics and computer vision, sparking broad research (Essakine et al., 2025). In particular, neural signed distance functions are often used to describe surfaces in three dimensions (Schirmer et al., 2024). Most work in this direction exclusively focuses on representing objects in three-dimensional space. However, we want to represent implicit submanifolds of arbitrary dimension and codimension. For this, signed distance functions are not a suitable implicit representation, and we will explore learning approximate projections instead in Section 4.

## 2. Background: Riemannian Geometry

In this paper, we consider an autoencoder for data $\mathcal{X} \subset \mathcal{M} \subset \mathbb{R}^n$ on a hidden manifold $\mathcal{M}$ consisting of an *encoder* $\phi \colon \mathbb{R}^n \to \mathbb{R}^l$ and *decoder* $\psi \colon \mathbb{R}^l \to \mathbb{R}^n$. We will describe the corresponding latent manifold $\mathcal{Z} := \phi(\mathcal{M})$ as an im-

plicit submanifold with a Riemannian metric. This enables us to develop suitable numerical schemes for geodesic interpolation and extrapolation without requiring an explicit parametrization. In contrast to implicit representations, global parametrizations of manifolds do not exist in general due to topological obstructions. Furthermore, learning a local parametrization of a manifold from point cloud data is a substantially harder problem.

Let us shortly recap the basics of the required Riemannian geometry. In Appendix D we provide a more intuitive introduction in a basic setting. We consider the manifold $\mathcal{Z} \subset \mathbb{R}^l$ as an implicit, $m$-dimensional submanifold, i.e. we have $\mathcal{Z} = \left\{ z \in \mathbb{R}^l \mid \zeta(z) = 0 \right\}$ for a smooth map $\zeta$. In our case,

$$\zeta \colon \mathbb{R}^l \to \mathbb{R}^l; \; \zeta(z) := z - \Pi(z), \tag{1}$$

where $\Pi \colon \mathbb{R}^l \to \mathbb{R}^l$ is the projection from the latent space $\mathbb{R}^l$ to the nearest point on the latent manifold $\mathcal{Z}$. The tangent space $T_z\mathcal{Z}$ at a point $z \in \mathcal{Z}$ is the vector space of all velocities of paths passing through $z$. For implicit submanifolds, $T_z\mathcal{Z} = \ker D\zeta(z)$. A *Riemannian metric* is an inner product $g_z(\cdot, \cdot)$ on $T_z\mathcal{Z}$ smoothly depending on $z$. This inner product allows to measure lengths of tangent vectors and angles between them. The simplest choice would be the Euclidean inner product inherited from $\mathbb{R}^l$, but it may be useful to apply other inner products that better represent the geometric structure of the original data.

The length of a path $\mathbf{z} \colon [0, 1] \to \mathcal{Z}$ with velocity $\dot{\mathbf{z}}$ is defined as $\mathcal{L}(\mathbf{z}) = \int_0^1 \sqrt{g_{\mathbf{z}(t)}(\dot{\mathbf{z}}(t), \dot{\mathbf{z}}(t))} \, \mathrm{d}t$, and the *Riemannian distance* $\mathrm{dist}(z_0, z_1)$ between two points $z_0, z_1 \in \mathcal{Z}$ is the infimum over all paths with endpoints $\mathbf{z}(0) = z_0$, $\mathbf{z}(1) = z_1$. A minimizing path is called a *geodesic*. A torus $\mathcal{Z}$ is shown as a toy example for $l = 3$ and $m = 2$ in Figure 3. A geodesic connecting $z_0$ and $z_1$ can equivalently be found by minimizing the path energy

$$\mathcal{E}(\mathbf{z}) = \int_0^1 g_{\mathbf{z}(t)}(\dot{\mathbf{z}}(t), \dot{\mathbf{z}}(t)) \, \mathrm{d}t \tag{2}$$

over curves $(\mathbf{z}(t))_{t \in [0,1]}$ in $\mathbb{R}^l$, subject to $\mathbf{z}(0) = z_0$, $\mathbf{z}(1) = z_1$ and $\zeta(\mathbf{z}) = 0$. Physically, geodesics have vanishing acceleration within the manifold, i.e. they always go straight at constant speed, neither changing direction nor velocity. Of course, viewed from the outside, a geodesic path on a curved manifold no longer looks straight. For the Euclidean inner product inherited from the ambient $\mathbb{R}^l$ as metric, the corresponding geodesic equation (the Euler–Lagrange equation associated with the minimization of $\mathcal{E}$) expresses the lack of acceleration *within* the manifold, $\ddot{\mathbf{z}}(t) \perp T_{\mathbf{z}(t)}\mathcal{Z}$. For implicit submanifolds, this reads as $D\zeta(\mathbf{z}(t))\ddot{\mathbf{z}}(t) = 0$ and $\ddot{\mathbf{z}}(t) \cdot w = 0$ for all $D\zeta(\mathbf{z}(t))w = 0$. For other Riemannian metrics, this geodesic ODE defining a geodesic for initial data $\mathbf{z}(0) = z$ and $\dot{\mathbf{z}}(0) = v \in T_z\mathcal{Z}$ becomes more complicated. Mapping $v$ to the arrival point

$y = \mathbf{z}(1)$ at time 1 yields the *Riemannian exponential map* $\exp_z \colon T_z \mathcal{Z} \to \mathcal{Z} \; : \; \exp_z v = y$.

## 3. Discrete geodesic calculus

We introduce a time-discrete geodesic calculus suitable for (imperfect) implicit manifold representations. A central ingredient is a local approximation $\mathcal{W}(\cdot, \cdot)$ of the squared Riemannian distance used to define the *discrete path energy*

$$\mathcal{E}^K(z_0, \ldots, z_K) = K \sum_{k=1,\ldots,K} \mathcal{W}(z_{k-1}, z_k) \quad (3)$$

of a discrete path $\mathbf{z} = (z_0, \ldots, z_K)$ as a discrete counterpart of (2). Consequently, a *discrete geodesic* for given endpoints $z_0, z_K \in \mathcal{Z}$ is a minimizer of (3) subject to the constraint $\zeta(z_k) = 0$ for $k = 0, \ldots, K$. Rumpf & Wirth (2015, Corollary 4.10) have shown that if $\mathcal{Z}$ is smooth, $z \mapsto g_z$ is Lipschitz continuous, and $\mathcal{W}(z_0, z_1)$ approximates $\mathrm{dist}(z_0, z_1)^2$ up to an error $O(\mathrm{dist}(z_0, z_1)^3)$, then the piecewise affine interpolations of minimizers of the discrete path energies $\mathcal{E}^K$ converge uniformly to minimizers of the continuous path energy $\mathcal{E}$. One can interpret $\mathcal{W}(z_{k-1}, z_k)$ physically as the energy of a spring connecting $z_{k-1}$ and $z_k$. Then, $\mathcal{E}^K(z_0, \ldots, z_K)$ is the total elastic energy of a chain of $K$ springs, which we relax under the constraint that all nodes lie on $\mathcal{Z}$. Depending on the application and the underlying configuration of the data manifold, one can distinguish different Riemannian metrics on $\mathcal{Z}$ and corresponding functionals $\mathcal{W}$:

- The Euclidean inner product $g_z(v, w) = v \cdot w$ inherited from the ambient space $\mathbb{R}^l$ leads to

$$\mathcal{W}_{\mathrm{E}}(z, \tilde{z}) = |\tilde{z} - z|^2. \quad (4)$$

- The *pullback metric* (in the case of an explicitly known metric on $\mathcal{M}$) $g_z(v, w) = g^{\mathcal{M}}_{\psi(z)}(D\psi(z)v, D\psi(z)w)$ with $g^{\mathcal{M}}_x \colon T_x \mathcal{M} \times T_x \mathcal{M} \to \mathbb{R}$ is reflected by

$$\mathcal{W}_{\mathcal{M}}(z, \tilde{z}) = \mathrm{dist}^2_{\mathcal{M}}(\psi(z), \psi(\tilde{z})). \quad (5)$$

- If $\mathcal{M}$ is equipped with the Euclidean inner product inherited from $\mathbb{R}^n$, $\mathcal{W}_{\mathcal{M}}$ simplifies to

$$\mathcal{W}_{\mathrm{PB}}(z, \tilde{z}) = |\psi(z) - \psi(\tilde{z})|^2. \quad (6)$$

We discuss this further in Appendix A.1.

**Computing discrete geodesics.** The Lagrangian for the constrained optimization problem of minimizing (3) subject to $\zeta(\mathbf{z}) = 0$ reads $\mathbf{L}(\mathbf{z}, \Lambda) = \mathcal{E}^K(\mathbf{z}) - \Lambda : \zeta(\mathbf{z})$, where $\mathbf{z} \in \mathbb{R}^{l,K+1}$ and $\Lambda \in \mathbb{R}^{l,K-1}$ denotes the matrix of the components of the Lagrange multiplier and $\Lambda : \zeta(\mathbf{z}) :=$

$\sum_{i=1}^l \sum_{k=1}^{K-1} \Lambda_{ik} \zeta_i(z_k)$. The optimality conditions for the constrained optimization can be expressed as the saddle point condition

$$\begin{aligned} 0 &= \partial_{z_k} \mathbf{L}(\mathbf{z}, \Lambda) \\ &= K \left( \partial_{z_k} \mathcal{W}(z_{k-1}, z_k) + \partial_{z_k} \mathcal{W}(z_k, z_{k+1}) \right) \\ &\quad - \sum_{i=1,\ldots,l} \Lambda_{ik} \nabla \zeta_i(z_k), \end{aligned} \quad (7)$$

$$0 = \partial_{\Lambda_{ik}} \mathbf{L}(\mathbf{z}, \Lambda) = -\zeta_i(z_k) \quad (8)$$

for $k = 1, \ldots, K-1$ and $i = 1, \ldots, l$.

We propose to use an augmented Lagrangian method to compute solutions of the constrained optimization problem. In detail, for the augmented Lagrangian

$$\begin{aligned} \mathbf{L}^a(\mathbf{z}_j, \Lambda_j, \mu_j) &= \mathbf{L}(\mathbf{z}_j, \Lambda_j) + \tfrac{\mu_j}{2} |\zeta(\mathbf{z}_j)|^2 \\ &= \mathcal{E}^K(\mathbf{z}_j) - \Lambda_j : \zeta(\mathbf{z}_j) + \tfrac{\mu_j}{2} |\zeta(\mathbf{z}_j)|^2 \end{aligned} \quad (9)$$

we iterate the update rules

$$\mathbf{z}_{j+1} = \operatorname*{arg\,min}_{\mathbf{z} \in \mathbb{R}^{l(K-1)}} \mathbf{L}^a(\mathbf{z}, \Lambda_j, \mu_j), \quad (10)$$

$$\Lambda_{j+1} = \Lambda_j - \mu_j \zeta(\mathbf{z}_{j+1}), \quad \mu^{j+1} = \alpha \, \mu^j \quad (11)$$

for given initial data $(\mathbf{z}_0, \Lambda_0, \mu_0)$ and some $\alpha > 1$. The update of the multiplier $\Lambda$ ensures that the Euler–Lagrange equation $\partial_{\mathbf{z}} \mathbf{L}^a = 0$ coincides with the first saddle point condition (7). The third term of $\mathbf{L}^a$ is a penalty ensuring closeness of $\mathbf{z}_j$ to $\mathcal{Z}$ and thus reflects the second saddle point condition (8).

The augmented Lagrangian approach is not harmed by the fact that the Lagrange multiplier in (7)-(8) is underdetermined (and thus nonunique) due to rank $D\zeta(z) = l - m$. Moreover, it is also used with inexact constraints (Frick et al., 2011; Jin, 2017), which is important since in practice we replace $\zeta$ with $\zeta_\sigma = \mathrm{id} - \Pi_\sigma$ for a learned *approximate* projection $\Pi_\sigma$ (cf. Section 4). The augmented Lagrangian method allows this approximation as long as $\zeta_\sigma$ points approximately in normal direction to $\mathcal{Z}$. Furthermore, the penalty term enforces small values of $\zeta_\sigma(z_k) = z_k - \Pi_\sigma(z_k)$ even though $\zeta_\sigma$ is not expected to vanish. Overall, we observe that the augmented Lagrangian method works for an inexact implicit function $\zeta_\sigma$ as long as $(\zeta_\sigma, D\zeta_\sigma)$ approximate $(\zeta, D\zeta)$ sufficiently well (see, e.g., Figure 3). We give details on the implementation and the parameters for the augmented Lagrangian approach in Appendix A.2.

**Computing a discrete exponential.** To derive a discrete counterpart of geodesic extrapolation via the exponential map, we aim at a numerical scheme, which reproduces discrete variational geodesics minimizing (3). To this end, we interpret the first saddle point condition

$$\partial_{z_k} \widehat{\mathbf{L}}((z_{k-1}, z_k, z_{k+1}), \lambda_k) := \partial_{z_k} \mathbf{L}(\mathbf{z}, \Lambda) = 0$$

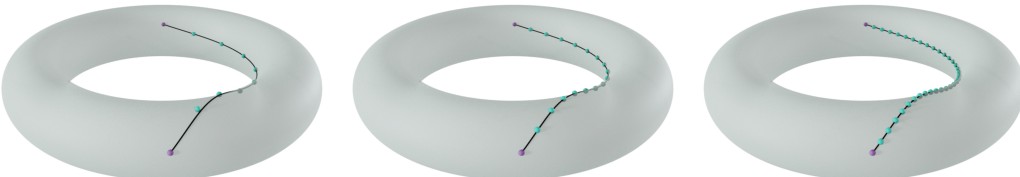

*Figure 3.* Discrete geodesics for different values of $K$ computed on a torus with learned implicit manifold representation $\zeta_\sigma$ (green points) and highly resolved geodesic computed with ground truth representation $\zeta$ (black line).

from the discrete geodesic interpolation (cf. (7)) as a (non-linear) equation in the unknowns $z_{k+1}$ and $\lambda_k$ for given $z_{k-1}, z_k$. Furthermore, we implement the constraint $\zeta(z_k) = 0$ from the second saddle point condition (8) as a penalty. This, leads to the minimization of the functional $\mathcal{F}: \mathbb{R}^l \times \mathbb{R}^l \to \mathbb{R}$,

$$\mathcal{F}(z_{k+1}, \lambda_k) := \tfrac{\mu}{2} |\zeta(z_{k+1})|^2 + \qquad (12)$$
$$|\partial_{z_k} \widehat{\mathbf{L}}((z_{k-1}, z_k, z_{k+1}), \lambda_k)|^2,$$

where $\mu > 0$ is some penalty parameter. As in the context of the discrete geodesic interpolation, this minimization remains reasonable if $\zeta$ is replaced by an approximation $\zeta_\sigma$. Now, given the first two points $z_0$ and $z_1$ and thus a corresponding discrete initial velocity $v_0 = K(z_1 - z_0)$, we iteratively minimize $\mathcal{F}(z_{k+1}, \lambda_k)$ for $k = 1, \ldots, K-1$ in both $z_{k+1}$ and $\lambda_k$ with a BFGS method. We define by $\mathrm{Exp}_{z_0}^K(v_0) := z_K$ the *discrete exponential map* as the final point of the extrapolated discrete geodesic $(z_0, \ldots, z_K)$ (cf. Figure 4). Following Rumpf & Wirth (2015, Theorem 5.10), the discrete exponential map converges to the continuous one as $K \to \infty$.

Functional (12) involves first derivatives of the distance approximation $\mathcal{W}$ and the implicit representation $\zeta$. However, these derivatives are not taken with respect to the optimization variable $z_{k+1}$. As a consequence, when computing the gradient of (12), the network representing $\zeta$ does not need to be differentiated twice, and only mixed second derivatives of $\mathcal{W}$ are required. Even when $\mathcal{W}$ is defined via a pullback metric, these derivatives can be computed without second-order differentiation of the decoder network.

## 4. Projection as implicit representation of latent manifolds

In Section 2, we defined the implicit function $\zeta(z) = z - \Pi(z)$ based on a projection $\Pi$ from the latent space to the latent manifold. In Section 3, we considered a yet unspecified approximation $\zeta_\sigma(z) = z - \Pi_\sigma(z)$. In this section, we will detail the concrete choice of $\zeta_\sigma$ or rather of the approximate projection $\Pi_\sigma$ as the minimizer of a suitable objective and discuss how to train its network representation.

**Projection as minimizer of a loss functional.** Let us suppose a density measure $\mathrm{d}z$ is given on the latent manifold $\mathcal{Z}$ reflecting the sampling on the data manifold. In applications, this is typically the empirical measure of the data samples or rather of their images under encoder $\phi$. The projection as a neural network is learned based on this possibly noisy point cloud representing the latent manifold. A natural approach is to learn it via the variational approach proposed by Alain & Bengio (2014) in the context of denoising autoencoders: We define the projection $\Pi_\sigma$ as the minimizer of the loss functional

$$\mathcal{Q}(\Pi) = \int_{\mathbb{R}^l} \int_{\mathcal{Z}} |z - \Pi(y)|^2 f_\sigma(z - y) \, \mathrm{d}z \, \mathrm{d}y \qquad (13)$$

over maps $\Pi : \mathbb{R}^l \to \mathbb{R}^l$. Here, $f_\sigma(y)$ is the normal distribution with mean 0 and standard deviation $\sigma$. The functional $\mathcal{Q}$ is a coercive quadratic form. Hence, the condition that $\partial_\Pi \mathcal{Q}(\Pi_\sigma)$ vanishes uniquely classifies the minimizer $\Pi_\sigma$ and leads to $0 = 2 \int_{\mathbb{R}^l} \int_{\mathcal{Z}} (\Pi_\sigma(y) - z) \vartheta(y) f_\sigma(z - y) \, \mathrm{d}z \, \mathrm{d}y$ for all $\vartheta \in C_c^\infty$. Thus, one obtains the approximate projection

$$\Pi_\sigma(y) = \left( \int_{\mathcal{Z}} f_\sigma(y - z) \, \mathrm{d}z \right)^{-1} \int_{\mathcal{Z}} z f_\sigma(y - z) \, \mathrm{d}z$$

of $y$ in the neighborhood of $\mathcal{Z}$ as a Gaussian-weighted $\mathcal{Z}$–barycenter (cf. Alain & Bengio (2014)). For $\mathcal{Z}$ being an affine subspace of $\mathbb{R}^l$, $\Pi_\sigma$ is indeed the orthogonal projection on $\mathcal{Z}$. In general, $\mathrm{id} - \Pi_\sigma$ does not necessarily vanish on $\mathcal{Z}$ but comes with a defect of order $O(\sigma^2)$ in the relative interior of smooth latent manifolds and $O(\sigma)$ close to the boundary. In Appendix A.3 we comment on its relation to the score function.

**Learning the projection on encoded samples.** To practically minimize the objective (13), we parameterize $\Pi_\sigma$ by a fully connected neural network with ELU activation functions (Clevert et al., 2015). We minimize the objective (13) using the Adam optimizer (Kingma & Ba, 2014).

In Appendix A.3, we study the properties of the denoising loss and the resulting $\Pi_\sigma$ on a low-dimensional toy model and show experimentally that the approximation error decreases for increasing point cloud size, increasing network architectures, and decreasing noise levels.

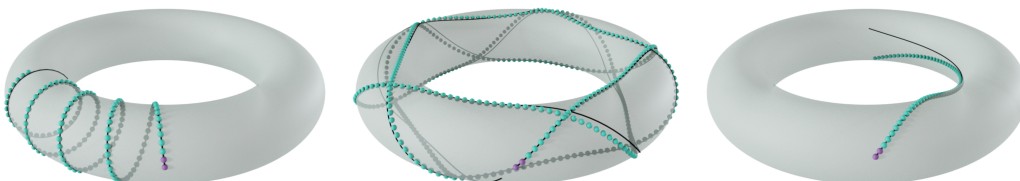

*Figure 4.* Comparison between computed exponentials with learned implicit manifold representation $\zeta_\sigma$ (green points) and ground truth representation $\zeta$ (black line). As in any dynamical system, slight numerical inaccuracies lead to an exponentially growing divergence (which is known to be more pronounced in regions of negative curvature as in the right-most example). See also the experiment in Figure 15.

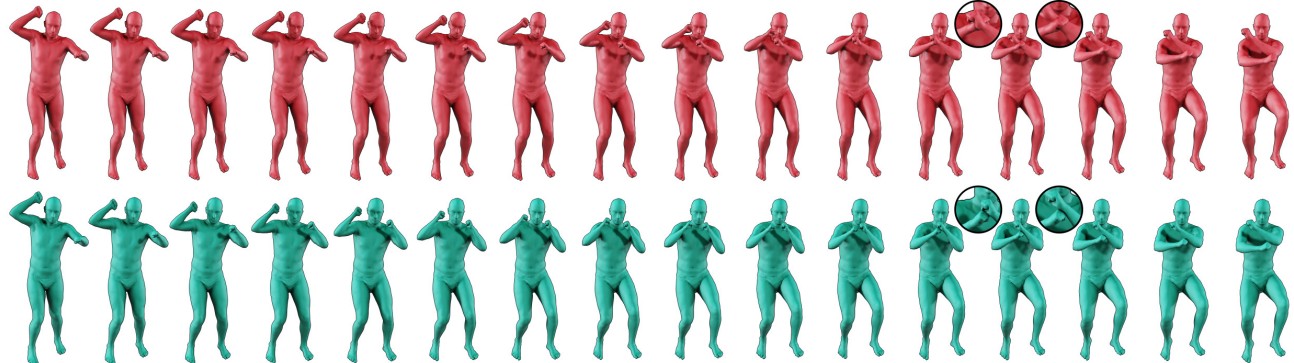

*Figure 5.* Interpolations on a learned submanifold of the shape space of discrete shells. Comparison between linear interpolation in latent space (red) and geodesic interpolation using a learned implicit representation (green) of the latent manifold $\mathcal{Z}$.

In applications, we have an approximation of the latent manifold $\mathcal{Z}$ by a point cloud $\phi(\mathcal{X})$ of encoded data samples $\mathcal{X}$. These point clouds can be sparse and noisy depending on the distribution of the data and the regularization of the autoencoder. Suitable values of $\sigma$ depend on the sampling. For data with low noise level and high point cloud density, small values of $\sigma$ are preferable as long as the convolution with the Gaussian $f_\sigma$ sufficiently regularizes the data distribution. On the other hand, a reliable projection further away from $\mathcal{Z}$ can only be expected for sufficiently large $\sigma$. In our experiments below, we could choose the same $\sigma$ for similar point cloud densities.

# 5. Results

In the following examples, we demonstrate the performance of the method across different types of data and latent manifolds obtained from different autoencoders.

## 5.1. Discrete shells / isometric autoencoder

We compute interpolations and extrapolations on a submanifold of the space of discrete shells (Grinspun et al., 2003), i.e., the space of all possible immersions of a fixed triangle mesh, and equip this space with the Riemannian metric proposed by Heeren et al. (2014). First, we train an isometric autoencoder to approximate this submanifold and then the projection operator as described in Section 4, allowing us to

perform the geodesic calculus on the latent manifold.

The submanifold is designed to approximate a dataset of shapes, such as different poses of a humanoid model. Because many datasets provide only a limited number of examples, we first apply a classical Riemannian construction to obtain the submanifold and its parametrization, which is computationally demanding. We then use this parametrization to create a denser training set for our autoencoder. We discard any pairs where at least one shape exhibits self-intersections to preserve physical plausibility. See Appendix B.1, for details on the data generation and the autoencoder training. In this way, the autoencoder learns a low-dimensional latent manifold approximating the shape space submanifold, enabling us to perform discrete geodesic operations in reduced dimensions. Figure 1 illustrates the overall procedure of our approach for an example of a two-dimensional submanifold of the manifold of discrete shells.

**Geodesic calculus on the latent manifold.** As described in Section 4, we use the trained projection operator $\Pi_\sigma$ to construct an approximate implicit representation of the latent manifold. For the geodesic calculus, we use the Euclidean metric, as this agrees with the shell distance due to the autoencoder's isometry. In Figure 5, we show a result of applying this approach to the SCAPE dataset (Anguelov et al., 2005) of human character poses and compare linear interpolation in latent space with geodesics on the latent

manifold computed using our approach. Our geodesics avoid self-intersections more effectively than linear interpolation, thanks to the rejection of intersecting shapes during training. Since our learned projection $\Pi_\sigma$ is only accurate up to the filter width $\sigma$, slight self-intersections may remain. We show additional examples in Appendix B.1.

### 5.2. Motion capture data / spherical variational autoencoder

We consider a latent manifold resulting from training a spherical variational autoencoder (SVAE) (Davidson et al., 2018) on motion capture data from the (CMU Graphics Lab). A suitable representation for the data was described by (Tournier et al., 2009): A pose can be represented as an element of $\mathrm{SO}(3)^m$, where $m = 30$ is the number of joints in the skeleton. Here, we identify rotation matrices with corresponding rotation vectors. The vector of rotations specifies the rotations of the joints. Given the connectivity and lengths of the skeletal segments, the full pose can be reconstructed. Hence, our input data $\mathcal{X}$ lies on a hidden manifold $\mathcal{M} \subset \mathrm{SO}(3)^m$. Arvanitidis et al. (2022) used an SVAE autoencoder to take the hyperspherical nature of this data into account. Unlike standard VAEs, which assume a Gaussian prior in the latent space, the SVAE uses von Mises–Fisher (vMF) distributions for regularization, which indeed enforces a hyperspherical latent geometry. Details on the employed data and training are provided in Appendix B.2. We use $l = 10$ latent dimensions. In the case of (S)VAEs, the encoder and decoder maps are not deterministic but parameterize distributions. We sample the latent manifold $\mathcal{Z}$ by sampling from the encoder distribution and obtain decoded points by sampling from the decoder. For simplicity, when learning $\Pi_\sigma$ and calculating discrete geodesics on $\mathcal{Z}$, we ignore the nondeterministic nature of the encoder and decoder and, by a slight abuse of notation, denote by $\psi(z) \in \mathrm{SO}(3)^m$ the mean of the distribution obtained from decoding $z$. Another approach would be to follow Arvanitidis et al. (2022) and incorporate the Kullback–Leibler divergence, see (15) in Appendix A.1 and Appendix C.3.

In Figure 6 (left), we show a projection of the sampled latent manifold (from which we learn $\Pi_\sigma$) onto the three most relevant dimensions obtained from a PCA.

**Geodesic calculus on the latent manifold.** The encoder embedding is not close to isometric. Hence, it is appropriate to equip the latent manifold with the pulled-back spherical distance

$$
\begin{aligned}
\mathcal{W}_\mathcal{M}(z, \tilde{z}) &= \mathrm{dist}^2_{\mathrm{SO}(3)}(\psi(z), \psi(\tilde{z})) \\
&= \sum_{i=1}^{m} |\arccos(\psi(z)_i \cdot \psi(\tilde{z})_i)|^2 . \quad (14)
\end{aligned}
$$

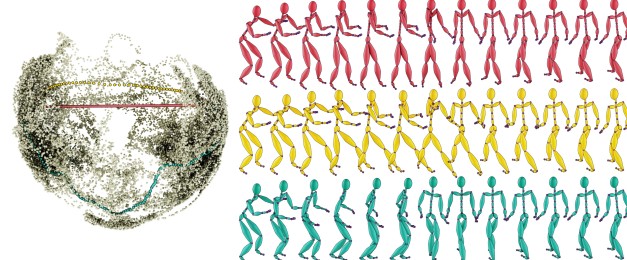

*Figure 6.* Left: Visualization of sample points in latent space (projected from $\mathbb{R}^{10}$ into $\mathbb{R}^3$ based on a PCA) and linear interpolation (red), geodesic interpolation with $\mathcal{W}_\mathrm{E}$ (yellow), and geodesic interpolation with $\mathcal{W}_\mathcal{M}$ (green). Right: Corresponding decoded sequences.

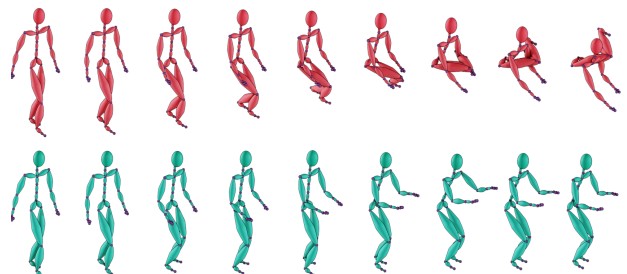

*Figure 7.* Decoded sequences starting from a fixed point in a fixed direction; linear extrapolation (red) and geodesic extrapolation with $\mathcal{W}_\mathcal{M}$ (green).

In Figure 6 (right), we compare geodesic interpolation based on $\mathcal{W}_\mathcal{M}$ (green) to geodesic interpolation with the Euclidean metric $\mathcal{W}_\mathrm{E}(z, \tilde{z}) = |z - \tilde{z}|^2$ (yellow) as well as linear interpolation in latent space (red). The results show that geodesic interpolation with the pullback metric yields realistic decoded paths, whereas linear interpolation in latent space leads to poses not lying on the data manifold $\mathcal{M}$, as indicated by unnatural contractions of shoulders and hips and by self-intersecting limbs. Figure 7 shows a pose extrapolation using the exponential map on the latent manifold. Different from linear extrapolation, the extrapolated path follows the geometry of the latent manifold and, after decoding, leads to realistic poses.

### 5.3. Image data / low bending, low distortion autoencoder

Next, we consider image data of a rotating three-dimensional object as proposed as an example by (Braunsmann et al., 2024). We use their regularized autoencoder, minimizing a loss function that promotes the embedding to be as isometric and as flat as possible. The data $\mathcal{X} \subset \mathcal{M} \subset \mathbb{R}^{128 \times 128 \times 3}$ consists of RGB images showing a toy cow model (Crane et al., 2013) from varying viewpoints. Training the regularized autoencoder requires computing distances and averages between dataset samples. Each image $x$ corresponds to a specific rotation $r_x \in \mathrm{SO}(3)$ which allows

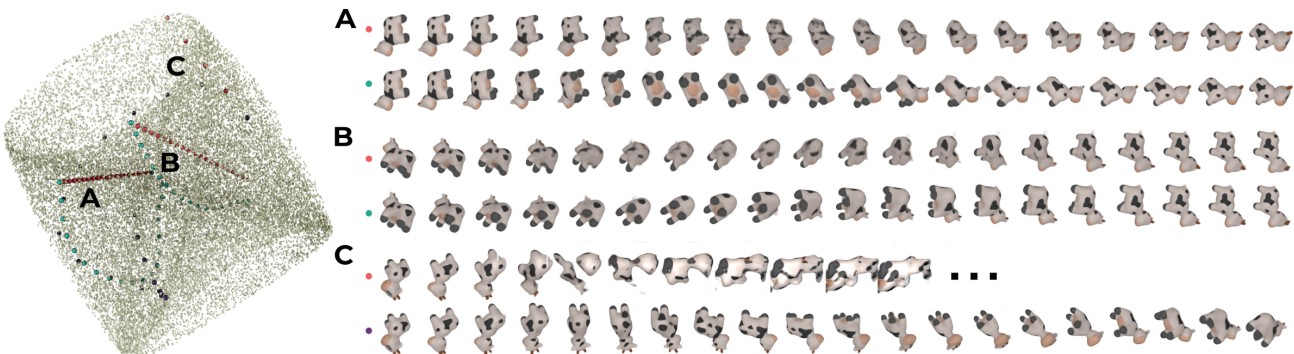

*Figure 8.* Left: Sample points in latent space (projected into $\mathbb{R}^3$; they represent an immersion of SO(3) which is topologically equivalent to the Klein bottle and thus has to self-intersect in three dimensions) and computed linear (red) and geodesic (green) interpolation (A, B) as well as linear (red) and geodesic (purple) extrapolation (C). Right: Corresponding decoded sequences.

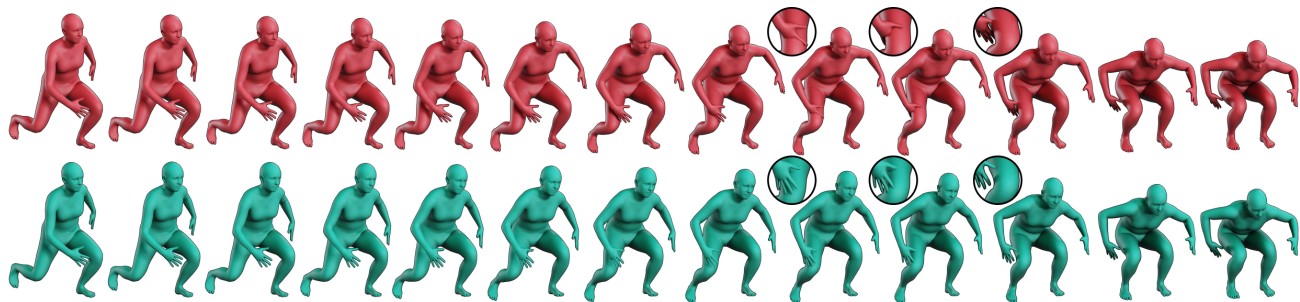

*Figure 9.* Linear interpolation in a quaternion representation of the SMPL body model (red) and geodesic interpolation on the manifold of plausible poses using the learned distance function provided by (Tiwari et al., 2022) (green).

to define these distances and averages. We use the publicly available pretrained autoencoder with $l = 16$-dimensional latent space, for details on other parameters and settings see Appendix B.3. A PCA projection on three dimensions of the resulting latent manifold is visualized in Figure 8 (left), the PCA analysis shows that the encoder uses six dimensions.

**Geodesic calculus on the latent manifold.** As the encoder is regularized to be near-isometric, we use $\mathcal{W}_E(z, \tilde{z}) = |z - \tilde{z}|^2$ in our geodesic calculus. In Figure 8, linear and geodesic interpolation as well as extrapolation in latent space are shown.

### 5.4. Extension to distance function as implicit representation

Finally, our discrete geodesic interpolation can be performed even if the latent manifold $\mathcal{Z}$ is merely represented by a distance function $d \colon \mathbb{R}^l \to \mathbb{R}$ with $\mathcal{Z} = \{z \in \mathbb{R}^l \mid d(z) = 0\}$. However, $d$ is non-differentiable on $\mathcal{Z}$. Hence, instead of the augmented Lagrange algorithm (10)-(11) we use the classical quadratic penalty method to minimize the discrete path energy $\mathcal{E}^K(\mathbf{z})$ subject to the constraints $d(z_k) = 0$. Computing the exponential map is not possible with missing normal information, though.

For example, Pose-NDF (Tiwari et al., 2022) provides a neural distance function to a manifold of plausible human poses. They use a quaternion representation of the SMPL body model by (Loper et al., 2015), resulting in $l = 84$ dimensions. Comparisons with linear interpolation in these coordinates show that our approach computes geodesics on the manifold of plausible poses, whereas linear interpolation produces self-intersecting poses that lie off the manifold, see Figure 9. An additional example is provided in Appendix B.4.

## 6. Limitations

Our experiments cover latent dimensions up to $l = 84$, and a thorough scalability analysis beyond this range has to be further explored. The per iteration cost of the augmented Lagrange optimization depends on the dimensionality of the unconstrained subproblem, however, such methods are well-suited for large-scale problems (Birgin & Martínez, 2014; Kanzow et al., 2018). The computation of the implicit representation is based on minimizing the denoising loss and thus is as dimension-dependent as general learning-based approaches. One advantage of our approach is that unlike other methods (Arvanitidis et al., 2016; 2018; 2022), we do not rely on latent space grids to compute geodesics, which

suffer from the curse of dimensionality.

We do not provide a full mathematical error analysis. This would require to control the error propagation combining the error of learning the implicit representation, which has an error of $O(\sigma^2)$ in an idealized setting, and the existing convergence results of the geodesic computation provided in (Rumpf & Wirth, 2015). We expect that the total error of geodesic computation could be bounded in terms of the input data and sampling noise, denoising parameter $\sigma$, and discretization parameter $K$.

A further limitation concerns the requirement for efficient local distance evaluation in the data manifold, or a near-isometric embedding thereof. In applications where such evaluation is infeasible, one may resort to a learned distance approximation, which we leave as a direction for future work.

## 7. Conclusion

Our results demonstrate that geometric operations on a latent manifold $\mathcal{Z}$ are indeed feasible. Central to our approach is a learned projection $\Pi_\sigma$ onto $\mathcal{Z}$.

Several directions emerge for future work. A natural step is to connect the projection-based representation via $\Pi_\sigma$ with distance-based representations $d$, for instance by deriving $\Pi_\sigma$ directly from $d$. From a numerical perspective, key open questions include how the accuracy of $\Pi_\sigma$ (and of the resulting geometric operators) depends on the sampling density and the scale parameter $\sigma$, how to enhance imperfect projections (e.g., using multiple or adaptive $\sigma$ or exploiting the property $\Pi_\sigma \approx \Pi_\sigma \circ \Pi_\sigma$), and how to improve augmented Lagrangian techniques for inexact constraints.

Conceptually, the next step lies in moving from latent manifolds to latent distributions, particularly in the context of VAEs and related generative models. Along these lines, an interesting possibility is to replace our projection operator with conditional denoisers, as used in diffusion models. Finally, extending the calculus to support detail transfer via discrete parallel transport and curvature approximation would open further applications.

## Acknowledgements

This work was supported by the Deutsche Forschungs-gemeinschaft (DFG, German Research Foundation) via project 211504053 – Collaborative Research Center 1060 and project 431460824 – Collaborative Research Center 1450 as well as via Germany's Excellence Strategy project 390685813 – Hausdorff Center for Mathematics and project 390685587 – Mathematics Münster: Dynamics–Geometry–Structure. Furthermore, this project has received funding from the European Union's Horizon 2020 research and innovation program under the Marie Skłodowska-Curie grant agreement No 101034255.

## Impact Statement

This paper presents work whose goal is to advance the field of Machine Learning. There are many potential societal consequences of our work, none which we feel must be specifically highlighted here.

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

# A. Method details

## A.1. Metric and Distance Approximation

We further discuss choices of the Riemannian metric and the corresponding functionals $\mathcal{W}$ to be used in our discrete geodesic calculus framework described in Section 3.

- The simplest choice for a metric on the latent manifold $\mathcal{Z}$ (though not particularly suitable for many applications) is the Euclidean inner product $g_z(v, w) = v \cdot w$ inherited from the ambient space $\mathbb{R}^l$. In this case,

$$\mathcal{W}_{\mathrm{E}}(z, \tilde{z}) = |\tilde{z} - z|^2$$

is the obvious choice fulfilling the requirements on $\mathcal{W}$. Physically, this choice corresponds to the energy of a single Hookean spring.

- If the data manifold $\mathcal{M}$ is embedded in $\mathbb{R}^n$ and equipped with the inherited Euclidean inner product, tangent vectors $v \in T_z\mathcal{Z} \subset \mathbb{R}^l$ correspond by the chain rule to embedded tangent vectors $D\psi(z)v \in \mathbb{R}^n$ after decoding. Hence the *pullback metric* (which gives tangent vectors to $\mathcal{Z}$ the same length as their counterparts on $\mathcal{M}$) is given by $g_z(v, w) = D\psi(z)v \cdot D\psi(z)w$. In this case, the squared Euclidean distance between the decoded points

$$\mathcal{W}_{\mathrm{PB}}(z, \tilde{z}) = |\psi(z) - \psi(\tilde{z})|^2$$

is an admissible approximation $\mathcal{W}$ of the squared Riemannian distance.

- Frequently, the data manifold $\mathcal{M}$ is equipped with a metric $g_x^{\mathcal{M}} : T_x\mathcal{M} \times T_x\mathcal{M} \to \mathbb{R}$. Again, pulling this metric back to $\mathcal{Z}$ yields

$$g_z(v, w) = g_{\psi(z)}^{\mathcal{M}}(D\psi(z)v, D\psi(z)w) = D\psi(z)^T G_{\psi(z)}^{\mathcal{M}} D\psi(z)v \cdot w,$$

where $G_x^{\mathcal{M}}$ is the matrix representation of the metric $g_x^{\mathcal{M}}$. In applications where the Riemannian distance on $(\mathcal{M}, g^{\mathcal{M}})$ can be explicitly computed, one is naturally led to

$$\mathcal{W}_{\mathcal{M}}(z, \tilde{z}) = \mathrm{dist}_{\mathcal{M}}^2(\psi(z), \psi(\tilde{z}))$$

as a proper choice for $\mathcal{W}$, measuring the squared Riemannian distance of decoded points $\psi(z)$ and $\psi(\tilde{z})$.

- In the case of non-deterministic decoders, $\psi(z)$ lies in a space of distributions. One can pull back the Fisher–Rao metric from the space of decoder distributions on the latent manifold. The Kullback–Leibler (KL)-divergence can then be used as a second-order distance approximation

$$\mathcal{W}_{\mathrm{KL}}(z, \tilde{z}) = \mathrm{KL}(\psi(z), \psi(\tilde{z})). \tag{15}$$

For a Gaussian decoder with fixed variances, where $\psi(z) = \mathcal{N}(\mu(z), \mathrm{I})$ is a normal distribution with mean $\mu(z)$, the KL-divergence reduces to the squared Euclidean distance of the means,

$$\mathrm{KL}(\psi(z), \psi(\tilde{z})) = \tfrac{1}{2}|\mu(z) - \mu(\tilde{z})|^2.$$

We refer to (Arvanitidis et al., 2022) for details on this information geometry perspective.

## A.2. Details on the Augmented Lagrangian method

In practice, we use a slightly more advanced version of the Augmented Lagrangian method following the algorithm described by Nocedal & Wright (2006, Chapter 17), which we adapt to our setting in Algorithm 1. Compared to the simplified version in the main text, we do not update the Lagrange multiplier and the penalty parameter in every iteration, but only depending on how well the constraint is already fulfilled. For the implementation of the minimization in (10) we implemented two versions, one using the LBFGS method from the PyTorch optimization package and a NumPy version using the BFGS method from SciPy (Virtanen et al., 2020). As the initial path we choose the path $\mathbf{z}_0 = \big(z_0 = z_0, \ldots, z_{\lfloor K/2 \rfloor} = z_0, \ z_{\lfloor K/2 \rfloor + 1} = z_K, \ldots, z_K = z_K\big)$ that remains constant and jumps directly from the given starting point $z_0$ to the endpoint $z_K$ at the middle time point. This is preferable to linear interpolation for initialization, as it ensures that the initial paths lies on or close to the manifold. We set the initial Lagrange multiplier $\Lambda_{i0} = 0$ for $i = 1, \ldots, K$ and $\alpha = 2$. The final tolerance $\eta^*$ for the constraint is problem-dependent and depends on the minimum values of $\zeta_\sigma$. In practice, a good rule-of-thumb is to choose $\eta^*$ as $K$ times the mean value on the embedded data samples $\eta^* \approx \frac{K}{|\mathcal{X}|} |\zeta_\sigma(\phi(\mathcal{X}))|$. The initial penalty parameter should be chosen sufficiently large to ensure that the first approximate solution remains close to the manifold.

---

**Algorithm 1** Augmented Lagrangian Method (Nocedal & Wright, 2006, Algorithm 17.4)

---

1: Choose initial point $\mathbf{z}_0$, multiplier $\Lambda_0$, penalty $\mu_0$, and $\alpha$
2: Choose final tolerances $\eta^*$ for constraint, $\omega^*$ for gradient, and maximum penalty $\mu_{\max}$
3: Set $\omega^0 \leftarrow 1/\mu^0$, $\eta^0 \leftarrow 1/\mu_0^{0.1}$
4: **for** $j = 0, 1, 2, \ldots$ **do**
5:     find approximate solution $\mathbf{z}_{j+1}$ of

$$\arg\min_{\mathbf{z}\in\mathbb{R}^{l(K-1)}} \mathbf{L}^a(\mathbf{z}, \Lambda_j, \mu_j)$$

6:     such that $|\nabla_{\mathbf{z}_{j+1}}\mathbf{L}^a(\mathbf{z}_{j+1}, \Lambda_j, \mu_j)| \leq \omega^k$
7:     **if** $|\zeta(\mathbf{z}_{j+1})| \leq \eta^k$ **then**
8:         **if** $|\zeta(\mathbf{z}_{j+1})| \leq \eta^*$ **and** $|\nabla_{\mathbf{z}_{j+1}}\mathbf{L}^a(\mathbf{z}_{j+1}, \Lambda_j, \mu_j)| \leq \omega^*$ **then**
9:             **return** $\mathbf{z}_{j+1}$ {final accuracy reached}
10:         **end if**
11:         update multiplier

$$\Lambda_{j+1} = \Lambda_j - \mu_j\zeta(\mathbf{z})$$

12:         update tolerances

$$\mu_{j+1} = \mu_j, \quad \eta_{j+1} = \eta_j/\mu_{j+1}^{0.9}, \quad \omega_{j+1} = \omega_j/\mu_{j+1}$$

13:     **else**
14:         increase penalty parameter

$$\mu_{j+1} \leftarrow \alpha\mu_j$$

15:         update tolerances

$$\Lambda_{j+1} = \Lambda_j, \quad \eta_{j+1} \leftarrow 1/\mu_{j+1}^{0.1}, \quad \omega_{j+1} \leftarrow 1/\mu_{j+1}$$

16:
17:         **if** $\mu_{j+1} > \mu_{\max}$ **then**
18:             **return** $\mathbf{z}_{j+1}$ {max penalty reached}
19:         **end if**
20:     **end if**
21: **end for**

---

### A.3. Details on learning the projection on encoded samples

We provide additional details on the learned projection (Section 4) used to construct an implicit representation of the latent manifolds.

**Optimization parameters.** We train a fully connected neural network with ELU activations (Clevert et al., 2015) by minimizing the loss functional (13) using the Adam optimizer (Kingma & Ba, 2014) with a learning rate of $10^{-3}$ and a weight decay of $10^{-5}$. The layer dimensions depend on the examples. The batch size to evaluate the integral over $\mathcal{Z}$ is 128 in all our examples. To approximate the inner integral, we generate a single sample $y_z = z + \epsilon$ where $\epsilon \sim f_\sigma$ for each data sample $z$ and optimization step.

**Parameter study.** To analyze the denoising loss and the resulting approximate projection $\Pi_\sigma$ for different parameters, we use the toy torus model to allow comparison with a ground truth projection and keep the evaluation visually tractable. In practice, only approximations of the latent manifold $\mathcal{Z}$ are available. To study the denoising property, we generate a noisy torus surface and train a projection with different values for $\sigma$, treating the noisy surface as $\mathcal{Z}$. We then visualize the image $\Pi_\sigma(\mathcal{Z})$ under the projections. As expected, a larger value of $\sigma$ leads to a stronger smoothing effect, see Figure 10.

For a point cloud without noise, a small parameter $\sigma$ leads to a higher accuracy of $\Pi_\sigma$ close to the surface but larger errors at certain distances, as those points are rarely seen in training, see Figure 11 (left). We further evaluate in Figure 11 the stability of the optimization, showing that the approximation error decreases for increasing point cloud size, increasing network architectures, and decreasing noise levels.

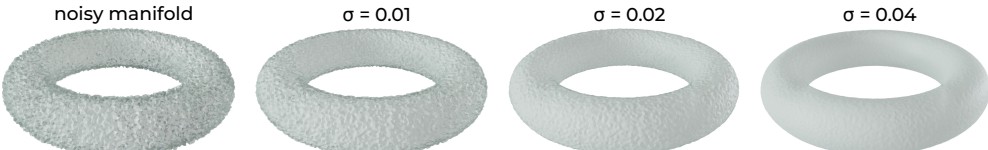

Figure 10. Visualization of the denoising effect for different choices of $\sigma$. Left: Noisy surface of unit diameter taken as $\mathcal{Z}$. Second left to right: Image $\Pi_\sigma(\mathcal{Z})$ under learned projections for different values of $\sigma \in \{0.01, 0.02, 0.04\}$. A larger choice of $\sigma$ leads to a projection onto a smoother surface.

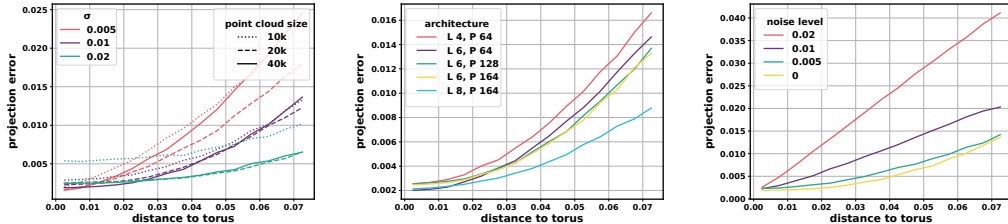

Figure 11. Error evaluation of learned torus projections versus distance to torus surface (which has unit outer radius). Left: Error for different sample sizes of torus and different values of $\sigma$; smaller $\sigma$ leads to a higher accuracy close to the surface and larger errors at certain distance. Middle: Error for different number of layers (L) and parameters per layer (P). Right: Error for training the projection on a torus surface with added Gaussian noise.

**Relation to score functions.** In this paragraph we briefly relate the denoising objective and its minimizer to approaches of diffusion models. A score function is the gradient of the log-probability density of a probability distribution $p$ with respect to the input: $\nabla_z \log p(z)$. Intuitively, it is a vector field that points in the direction of steepest increase in data likelihood at any given point $z$, it is therefor directly related to a projection onto the distribution (Kharitenko et al., 2026). If we now take a different perspective on the latent space and consider the latent samples $\phi(\mathcal{X}) \subset \mathbb{R}^l$ as samples of an unknown distribution $p$ it is shown in (Alain & Bengio, 2014, Thm. 2) that for $\sigma^2 \to 0$

$$\Pi_\sigma(z) = z + \sigma^2 \nabla \log p(z) + o(\sigma^2).$$

Hence, $\sigma^{-2}(\Pi_\sigma(z) - z)$ provides an approximation of the score function. In this setting the vector field $\Pi_\sigma(z) - z$ points towards high-density peaks of the latent sample distribution. Assuming for simplicity that the data is Gaussian distributed on the manifold $\mathcal{Z}$, i.e., with density $p(z) = \exp(-\operatorname{dist}^2(z, \mathcal{Z})/2\sigma^2)$, then the score $\sigma^2 \nabla \log(p(z))$ equals exactly $\Pi(z) - z$ for the projection $\Pi$ onto the manifold $\mathcal{Z}$. Today, score functions are at the heart of generative models and diffusion models (Rombach et al., 2022; Song et al., 2021), where the score of the data distribution is learned at different noise levels instead of fixing a noise parameter $\sigma$, and generation is then performed by following these score estimates backwards through a denoising process.

## B. Experiment details

### B.1. Details: Discrete shells / isometric autoencoder

We provide additional details on how we learn a latent manifold representing a submanifold of the shape space of discrete shells for the results given in Section 5.1.

**Data.** In this example, we use the SCAPE dataset (Anguelov et al., 2005) consisting of 71 immersions of a triangle mesh with 12500 vertices. To speed up the numerical algorithms used to create the samples for our autoencoder training, we reduced the resolution of the mesh using an iterative edge collapse approach to 1250 vertices. For visualization, we prolongated our results from the coarse to the fine mesh using a representation of the fine mesh vertices in terms of intrinsic positions and normal displacement with respect to the coarse mesh.

**Constructing the submanifold $\mathcal{M}$.** We begin by performing Principal Geodesic Analysis (PGA; (Fletcher et al., 2004)) on the input dataset. This involves three steps: (i) computing the Riemannian center of mass (the mean shape), (ii) mapping each input shape to the tangent space at the mean via the discrete Riemannian logarithm, and (iii) applying Principal

Component Analysis (PCA) to the resulting tangent vectors to obtain a low-dimensional linear subspace. The nonlinear submanifold is then recovered by applying the exponential map to this linear subspace.

All computations in this stage follow established numerical algorithms: For the Riemannian center of mass, the logarithms, and the exponential map, we use the methods introduced by (Heeren et al., 2014) with time resolution $K = 8$. For the tangent PCA, we use the representation using edge lengths and dihedral angles as described by (Sassen et al., 2020). We used the first two components for the example in Figure 1 and the first ten components for the example in Figures 5 and 12. To this end, we employed their publicly available C++ implementation (Heeren & Sassen, 2020).

**Learning the submanifold.** The training objective combines the reconstruction loss with the isometry loss introduced by Braunsmann et al. (2021) (see below). To generate training samples $\mathcal{X}$, we proceed as follows: First, we uniformly sample points within a hyperball of the linear tangent subspace. The size of the hyperball was chosen based on the norms of the projections of the input Riemannian logarithms onto the subspace. Second, for each such point, we sample a nearby point using a normal distribution with small variance centered on the first point. Finally, we apply the discrete exponential map to the two points to obtain a pair of points on the submanifold. We discard any pairs where at least one shape exhibits self-intersections to preserve physical plausibility. For the isometry loss, we compute the distance between the two shapes in a pair by computing discrete geodesics and taking their length. For the example in Figure 1, we drew 10000 pairs this way, and, for the example in Figures 5 and 12, we drew 100000 pairs. All these computations were performed with the same setup as described above.

The autoencoder architecture is a fully connected network with ELU activations. The encoder and decoder each have five layers, with the encoder reducing the input dimension from 3750 (three times the number of vertices) to a latent dimension of 24 and the decoder expanding it back accordingly.

**Projection learning.** We learn the projection on the embedded samples using $\sigma = 0.05$, six fully connected layers with 128 intermediate dimensions, and ELU activations.

In Figure 12, we provide additional comparisons between linear interpolation in the latent space and geodesic interpolation using our learned implicit representation.

### B.2. Details: Motion capture data / spherical variational autoencoder

We provide additional details for the motion capture experiment described in Section 5.2.

**Data.** We use the sequences of subject 86 trial 1-6 from the CMU Graphics Lab Motion Capture Database (CMU Graphics Lab). These are approximately 52000 frames. We define a pose as an element of $\mathrm{SO}(3)^m$ as described in the main text and transform the data to this representation using an AMC parser (Zhou). We take $80\,\%$ as training data and $20\,\%$ for testing.

**SVAE network.** We use the pythae framework (Chadebec et al., 2022) for implementing an *SVAE* network with a decoder that decodes to vMF distributions without fixed variances. We use $l = 10$ latent dimensions. We optimize with AdamW (Loshchilov & Hutter, 2019) using a batch size of 100, an initial learning rate of $10^{-3}$, and an adaptive learning rate scheduler with a patience of 10 and reduce by a factor of 0.05. We use two fully connected layers to learn the embedding with dimensions (90, 30, 10) and a separate second layer (30, 1) for the variance. For the decoder, we have dimension (10, 30, 90) followed by one layer (90, 90) for the decoded mean and one layer (90, 30) for the decoded variances.

**Projection learning.** We train the projection on the embedded samples by embedding the full set of training data and sampling one point per resulting vMF distribution. We use a fully connected network with 4 layers of 64 intermediate dimensions and Leaky ReLU activation functions, and we choose $\sigma = 0.05$.

In Figure 13, we provide an additional example of geodesic interpolation. Moreover, as further variant, in this figure we also show a path computed using the pullback metric $\mathcal{W}_{\mathcal{M}}$ but without using the implicit representation $\zeta_\sigma$ as constraint.

### B.3. Details: Image data / low bending, low distortion autoencoder

We provide additional details for the experiment described in Section 5.3.

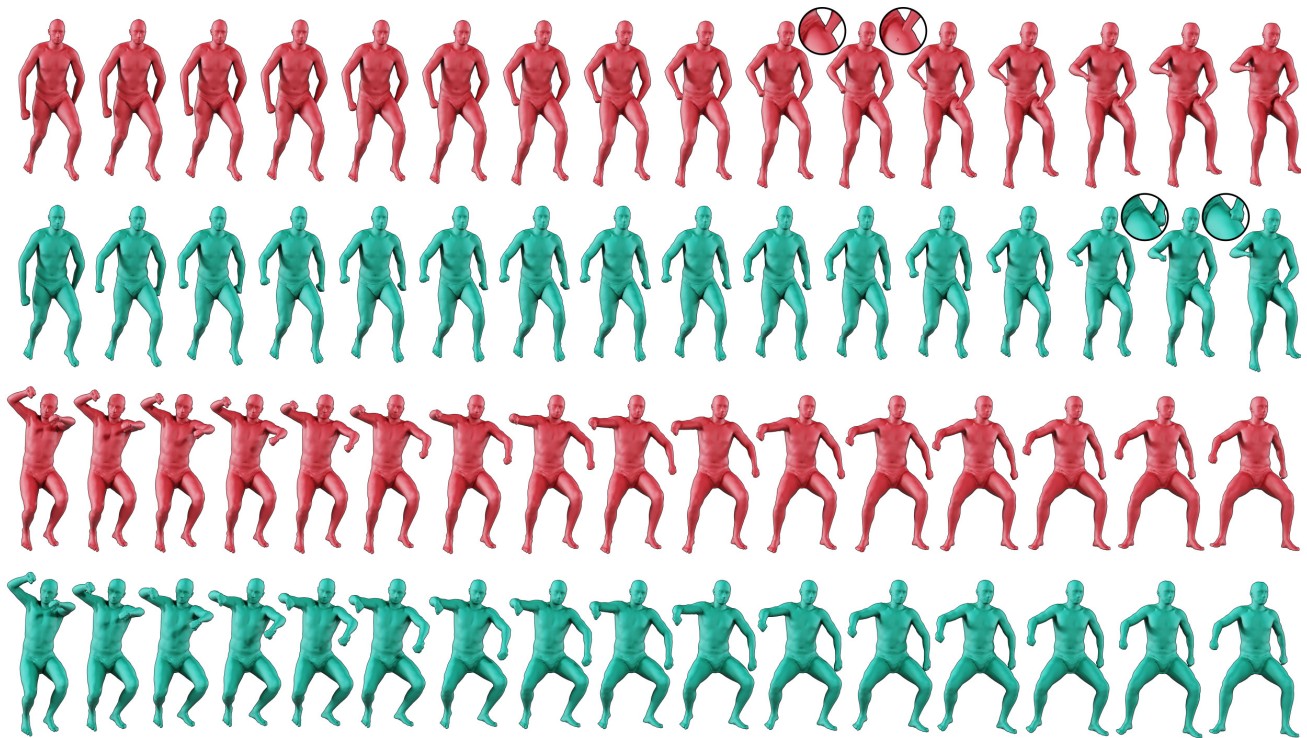

*Figure 12.* Two comparisons between linear interpolation (red) and geodesic interpolation using a learned projection on the embedded submanifold of discrete shells (green). Away from self intersections linear and geodesic interpolation are qualitatively similar indicating that the manifold is flat in these regions (bottom).

**Data.**  We use the code provided by (Braunsmann et al., 2024) to generate 30000 colored images with resolution $128 \times 128$ showing random rotations of the cow model.

**Low bending and low distortion autoencoder.**  Each image $x$ corresponds to a specific rotation vector $r_x$. Hence, a distance between the images can be defined as $\text{dist}_{\mathcal{M}}(x, y) = \arccos(r_x \cdot r_y)$ and geodesic averages $\text{av}_{\mathcal{M}}(x, y)$ as renderings of the object with the mean rotation between $r_x$ and $r_y$. The autoencoder is trained with tuples of nearby points $(x, y) \in \mathcal{X}_\epsilon$, where $\mathcal{X}_\epsilon \subset \{(x, y) \in \mathcal{M} \times \mathcal{M} \mid \text{dist}_{\mathcal{M}}(x, y) \leq \epsilon\}$. The regularization loss $\mathcal{J}_{\text{reg}}$ for the encoder is given by

$$\mathcal{J}_{\text{reg}}(\phi) = \frac{1}{|\mathcal{X}_\epsilon|} \sum_{x,y \in \mathcal{X}_\epsilon} \gamma\left(\frac{|\phi(x) - \phi(y)|}{\text{dist}_{\mathcal{M}}(x, y)}\right) + \lambda \frac{|\phi(\text{av}_{\mathcal{M}}(x, y)) - \text{av}_{\mathbb{R}^l}(\phi(x), \phi(y))|^2}{\text{dist}_{\mathcal{M}}(x, y)^4},$$

where $\gamma(s) = |s|^2 + |s|^{-2} - 2, \text{av}_{\mathbb{R}^l}(a, b) = (a + b)/2$ denotes the linear average, and $\lambda > 0$. The first term promotes an isometric embedding, encouraging $|\phi(x) - \phi(y)| = \text{dist}_{\mathcal{M}}(x, y)$. The second term penalizes the deviation between the embedding of the manifold average and the Euclidean average of the embedded points, favoring a flat embedding. For details, we refer to (Braunsmann et al., 2024).

We use the publicly available pretrained model for flatness weight $\lambda = 10$ and $l = 16$ latent dimensions.

**Projection learning.**  We learn the projection on the embedded samples using $\sigma = 0.005$, eight fully connected layers with 128 intermediate dimensions, and ELU activation functions.

### B.4. Details: Pose-NDF

Figure 14 shows an additional example using the neural distance function Pose-NDF (Tiwari et al., 2022) as manifold representation, see Section 5.4. We compare linear interpolation in the quaternion representation of the SMPL body model used in (Tiwari et al., 2022) and geodesic interpolation on the manifold corresponding to the approximate zero-level set.

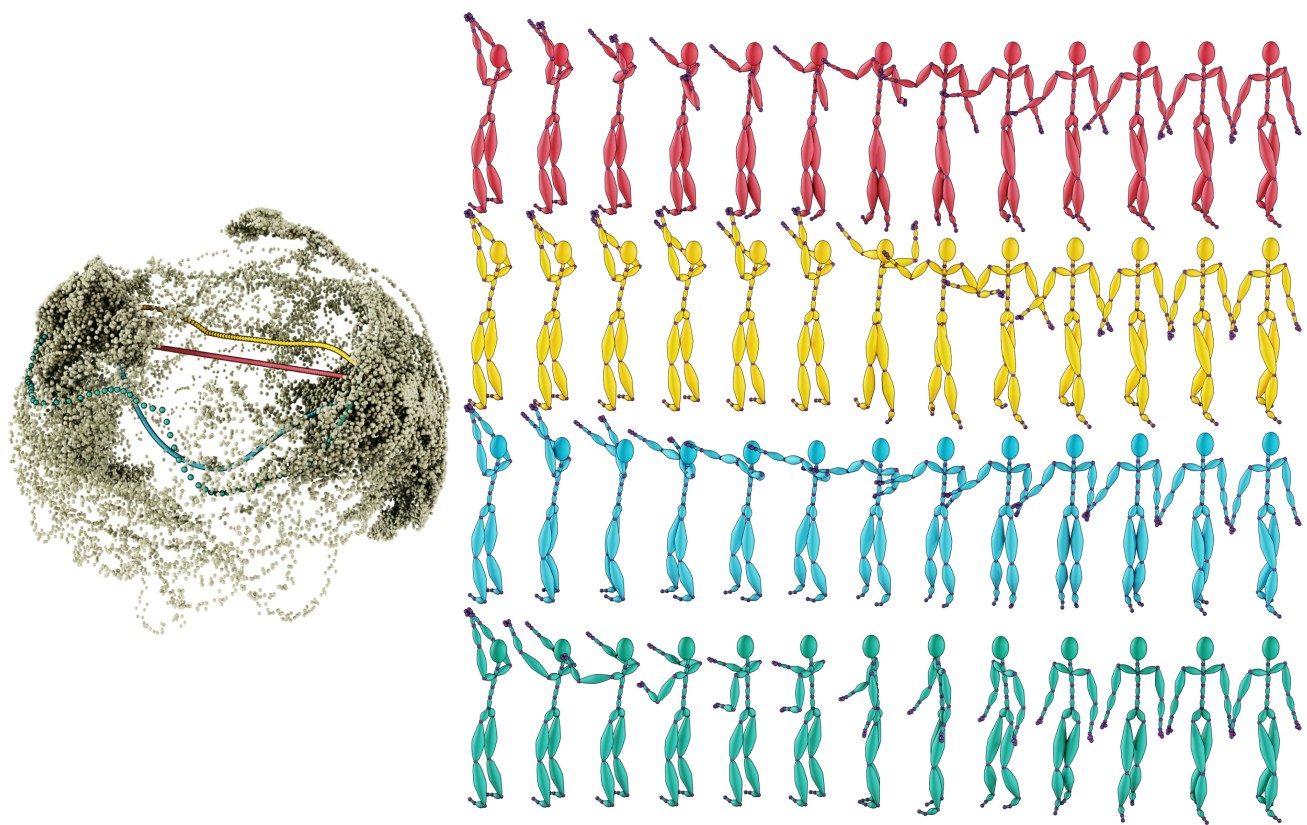

*Figure 13.* Same as Figure 6 for a different pair of endpoints. Left: Visualization of sample points in latent space (projected from $\mathbb{R}^{10}$ into $\mathbb{R}^3$) and computed paths between two points via linear interpolation (red), geodesic interpolation with $\mathcal{W}_E$ (yellow), unconstrained interpolation with $\mathcal{W}_\mathcal{M}$ (blue), and geodesic interpolation with $\mathcal{W}_\mathcal{M}$ (green). Right: Corresponding decoded sequences.

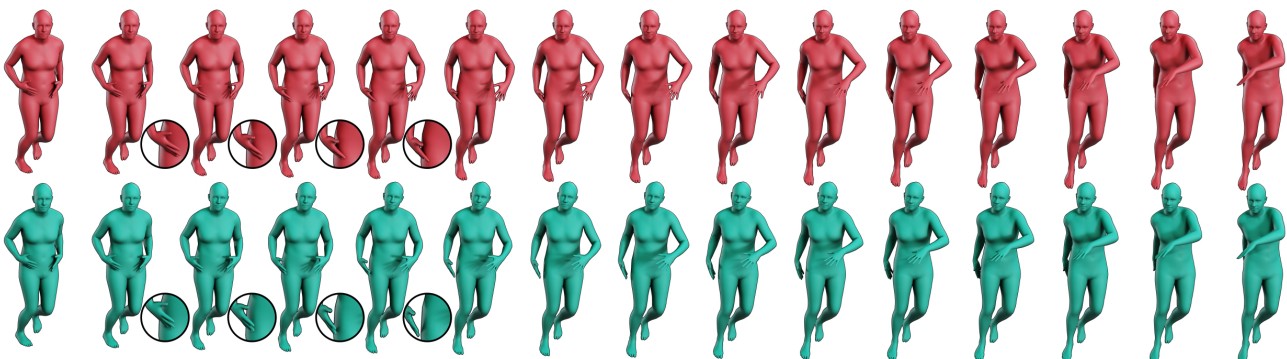

*Figure 14.* Linear interpolation in a quaternion representation of the SMPL body model (red) and geodesic interpolation using the neural distance function Pose-NDF to a manifold of plausible poses (green).

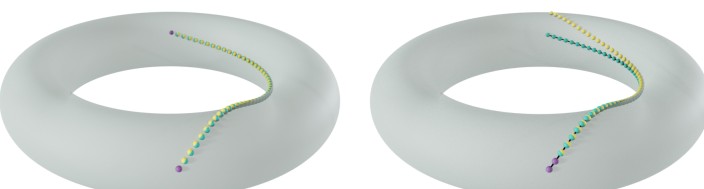

*Figure 15.* Left: geodesics computed with our approach (green) and the Stochman library (yellow), right: exponential maps for the initial direction taken from our discrete path energy minimizing geodesic shown for our approach (green) and for the Stochman library tool (yellow).

## C. Comparisons

In this section, we compare different numerical approaches to compute geodesics in settings related to the one considered in this article.

### C.1. Comparison to a parametric viewpoint and different discretization

A parametric viewpoint represents a manifold as the image of a coordinate map from a lower-dimensional parameter domain. This viewpoint is taken in many numerical approaches such as the well-established Stochman library (Detlefsen et al., 2021). In contrast, we take the viewpoint of an implicit manifold defined as the zero level set of a function in an ambient (latent) space. In both cases, geodesics between fixed endpoints are computed as solutions of a variational problem: minimizing the path energy. In our setting, this minimization is subject to the constraint to stay on the zero level set defined in (1) and is incorporated in the time-discrete scheme as described in Section 3. The exponential map (i.e. the geodesic for a prescribed initial point and velocity) corresponds to the solution of an initial value problem for a second-order ordinary differential equation. Instead of employing off-the-shelf PyTorch-based ODE solvers as in the Stochman library, we use a variational ODE solver, which ensures that the discrete path computed by the discrete exponential map well approximates a discrete geodesic between its start and end point. As a toy model, we compute geodesics on the torus $\mathcal{M} = \{z = (z^1, z^2, z^3) \in \mathbb{R}^3 \mid 0 = \zeta(z) := (\sqrt{(z^1)^2 + (z^2)^2} - R)^2 + (z^3)^2 - r^2\}$. In our approach, we minimize the discrete path energy

$$\mathcal{E}^K(z_0, \ldots, z_K) = K \sum_{k=1,\ldots,K} \|z_k - z_{k-1}\|^2 \tag{16}$$

for prescribed endpoints and under the constraint $\zeta(z_k) = 0$ for all $k = 0, \ldots, K$. For the parametric approach, we parametrize the torus by $[0, 2\pi]^2 \ni (\theta_1, \theta_2) \mapsto ((R + r\cos\theta_2)\cos\theta_1, (R + r\cos\theta_2)\sin\theta_1, r\sin\theta_2) \in \mathbb{R}^3$, which yields the metric tensor

$$G(\theta_1, \theta_2) = \begin{pmatrix} (R + r\cos\theta_2)^2 & 0 \\ 0 & r^2 \end{pmatrix}.$$

Using the Stochman library (Detlefsen et al., 2021), the curve energy induced by this metric is minimized using a cubic spline approximation evaluated at discrete timesteps on the two-dimensional parameter domain. Using our approach with 50 nodes ($K = 49$), the minimization takes 1.89 seconds and yields an approximated geodesic with path energy 4.7889. To reach comparable accuracy with Stochman, we use a spline with 8 nodes and 50 metric evaluation points and apply 2000 iterations for the optimization, resulting in a path energy of 4.7892. The minimization takes 2.48 seconds and no qualitative difference between the different discrete geodesics is visible; see Figure 15 (left).

Using our variational discretization for the exponential map (see Section 3) and considering the first two points of the discrete variational geodesic to define the initial direction, we obtain a discrete exponential map with 50 timesteps in less than a second. It reproduces the discrete variational geodesic; see Figure 15 (right). Using the tool for the exponential map in the Stochman library, the computation also takes less than one second. However, the result deviates visibly from that of our variational scheme, which is a manifestation of the robustness of our scheme and points to the moderate instability observed when computing exponential maps in negatively curved spaces. In applications to real data sets, we consider robustness to be particularly desirable. Finally, note that our approach of discretizing geodesics is not restricted to implicit manifolds; in fact, it was originally formulated for a parametric viewpoint (Rumpf & Wirth, 2015).

## C.2. Comparison of numerical methods for geodesics on sampled manifolds

In the setting of latent manifolds, one does not have direct access to a continuous metric tensor or a parameterized surface representation, but only to a discrete manifold representation given by samples. Several approaches have been proposed to compute geodesics in this setting. Some methods treat the manifold as a point cloud and rely on graph-based shortest-path algorithms (Sorrenson et al., 2025; Mckenzie & Damelin, 2019). However, such approaches are generally not suited for computing an exponential map. Arvanitidis et al. (2022) precompute curve energies on a uniform grid in latent space, use graph-based shortest-path methods to obtain an initial geodesic, and subsequently refine it via cubic-spline optimization. This grid-based construction does not scale well with increasing latent dimension, and special treatment is required to connect the start and end points to the computed geodesic. In (Arvanitidis et al., 2019), the authors discuss ill-conditioning and failure modes of standard solvers for computing geodesics on learned manifolds. As an alternative, they propose a fixed-point iteration scheme that avoids explicit Jacobian evaluations. A specific discretization of the exponential map is not discussed. In (Shao et al., 2018), the exponential map is approximated via parallel transport. We will revisit several of these methods in the comparisons presented in Appendix C.3 and Appendix C.4.

## C.3. Comparison to uncertainty-regularized variational decoders

In (Arvanitidis et al., 2022), the Kullback–Leibler (KL) divergence or equivalently its second-order approximation (15), is used as an approximation of the Fisher information metric on the latent manifold of variational autoencoders (VAEs). To compute geodesics on the latent manifold, this metric is not sufficient. Indeed, information on the actual geometry of the latent manifold as a submanifold of the latent space is needed. As a remedy, Arvanitidis et al. (2022) extend the metric to be very costly away from the latent manifold, such that shortest paths avoid crossing this outside region. In principle, this extension can be done in different ways; Arvanitidis et al. (2022) choose to decode points off the latent manifold as probability distributions with a much larger variance, dubbed "uncertainty regularization", and then use the above-described KL divergence-based Riemannian metric on the full latent space. The authors calibrate the decoder in a post-processing step via an estimation of distances of latent points to the latent data manifold. They approximate these distances via Euclidean K-means clustering and choose the distance to the nearest cluster center (Arvanitidis et al., 2022)[Appendix C.2]. Based on these distances, they apply a sigmoid function to interpolate between the decoder variance and a large fallback variance. Then, geodesics can be computed either by first discretizing the manifold on a grid in latent space, as described in (Arvanitidis et al., 2022), or by directly minimizing the curve energy. Note that, just like in our method, this approach requires establishing a representation of the latent manifold, i.e., a distance function, whereas we use a projection operator onto the latent manifold. The subsequent translation of the distance function into a Riemannian metric on the full latent space is solely for the purpose of computing geodesics via unconstrained minimization. Then, geodesics are actually allowed to leave the latent manifold and will typically only stay in its vicinity. Thus, it is conceptually cleaner to directly compute geodesics constrained to the latent manifold, which is the approach we pursue. Furthermore, a distance function is in general less robust to compute than a projection: while the former is nonsmooth for latent manifold generalization with locally varying codimension, the latter is not. Finally, finding the nearest cluster center for each point in latent space to approximate the distance function does not scale well with latent manifold dimension, which is why, in our approach, we instead leverage the power of deep learning to learn a projection.

To compute geodesics, a pullback metric based on the mean and variance of the decoder is used in (Arvanitidis et al., 2018), together with an extension to the full latent space using a regularized decoder variance that is large away from the data support. For training, a radial-basis-function network (RBF) (Broomhead & Lowe, 1988) is employed. Furthermore, a Euclidean $k$-nearest-neighbor graph with edge weights given by Riemannian distances is constructed. Then, at query time, the endpoints are projected onto the graph, a discrete shortest path is extracted, and the resulting polyline is smoothed and interpolated with a time-parameterized cubic spline to obtain an initial continuous geodesic approximation.

A comparison of those two approaches with our method is given in Figure 16 for a toy distribution. To this end, we train a VAE that decodes into normal distributions with different means and variances and learn the projection onto the latent manifold, which takes approximately 12 minutes. Based on this, we compute geodesics with our method using the Euclidean metric in latent space (4 seconds), the pullback of the Euclidean metric of decoded samples (30 seconds), and for the KL-divergence-based metric (1 minute). For the algorithm accompanying (Arvanitidis et al., 2022), the calibration with sigmoid functions takes 5 seconds, and the minimization of the path energy takes 10 seconds. However, the resulting geodesic leaves the data manifold, and the optimization does not converge. In the framework of (Arvanitidis et al., 2018) with the radial-basis-function network, the accompanying code takes 21 seconds for calibration and 45 seconds to compute geodesics using the pullback of the mean and variance. While for low-dimensional latent manifolds the calibration of the

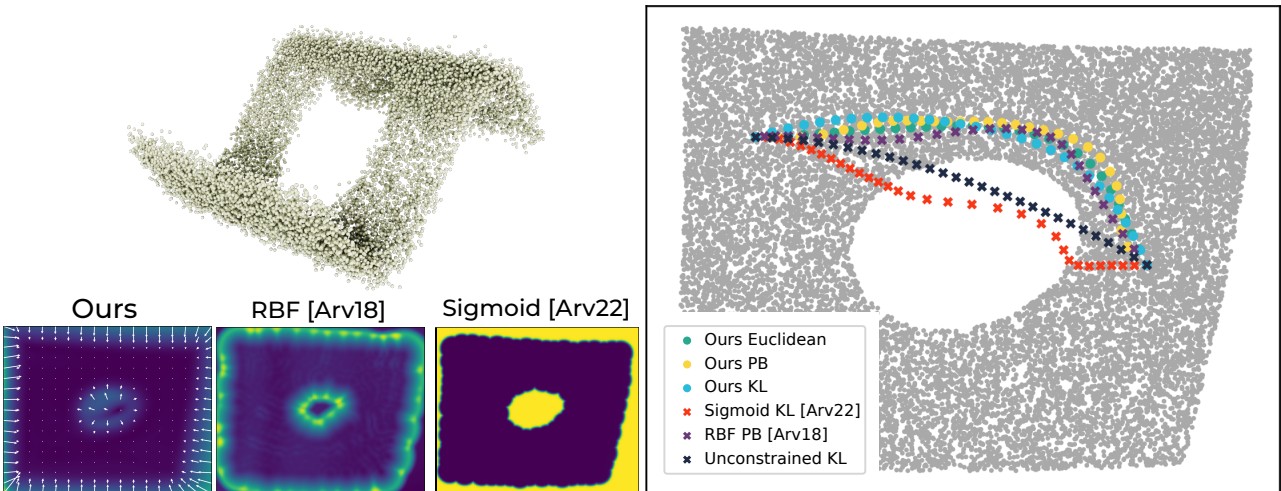

*Figure 16.* Top left: 3D input point cloud; bottom left: our learned representation of the 2D learned latent manifold and a visualization of the calibrated covariances using the respective method; right: latent manifold with geodesics computed using different methods.

decoder is faster than learning our projection, these approaches are limited to VAEs and to models without fixed decoder variance. In general, the optimization is harder, together with the previously mentioned conceptual disadvantages.

### C.4. Comparison to density based metrics

There is a line of work that aims at defining density-based metrics on a data distribution that are cheap in areas of high data density and expensive otherwise. This allows to compute trajectories that follow the data distributions. Examples include Fermat distances (Trillos et al., 2024), estimating local covariance matrices by fitting normal distributions (Arvanitidis et al., 2016), and score-based approaches in the context of diffusion models (Yu et al., 2025; Lobashev et al., 2025).

We compare our method with an approach that fits locally adaptive normal distributions (LAND model) (Arvanitidis et al., 2016), implemented using the Stochman library (Detlefsen et al., 2021). Rather than fitting full normal distributions, the method restricts itself to computing local samples with diagonal covariance matrices for the local data distribution around each point, and then uses the associated Mahalanobis inner product as a Riemannian metric, weighted by the inverse data point density. We consider two settings: first, the original application to point-cloud-represented data distributions, and second, a latent manifold example. Let us remark that although our method is designed for latent manifolds, it can be applied in both scenarios.

First, we show a comparison on a toy data distribution in Figure 17, where the data consist of a noisy sampling of a half-circle. Shortest path computations lead to qualitatively similar results for our approach and for the LAND model. We observe that our method performs better in the case of a larger distance between the chosen endpoints and leads to a more stable exponential computation.

In principle, the LAND model can be used in latent spaces with custom metrics different from the Euclidean one. However, as reported in (Arvanitidis et al., 2022), this is computationally demanding, as it requires repeated computations of Riemannian exponential maps and logarithms. To cope with this drawback, the authors propose to precompute these maps on a grid in latent space, which is naturally restricted to low-dimensional latent spaces. For a comparison with our method, we therefore consider the example of the close-to-isometric embedding from Section 5.3 with a low-dimensional latent space in order to be able to directly apply the method with the Euclidean metric; see Figure 18. The computed geodesics of the LAND model are of acceptable quality but show some artifacts, indicating that the discrete geodesic paths leave the neighborhood of the latent manifold. In fact, we emphasize that the model is designed for volumetric data of varying density rather than for manifolds with non-vanishing codimension.

For the experiment in Figure 17, we use $\sigma = 0.04$ for learning the projection, a network with four fully connected layers, 64 intermediate dimensions, and ELU activation functions. The dataset contains 6000 points. Training up to the selected epoch takes approximately ten minutes. Computing all five geodesics with $K = 30$ requires less than ten seconds, while the

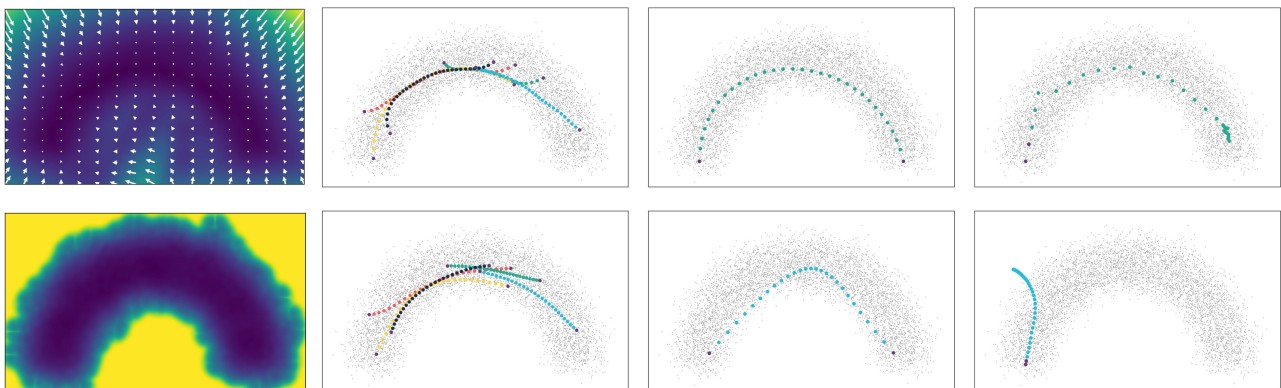

*Figure 17.* Top row from left to right: level sets of our learned representation function, five geodesics with randomly sampled start and end points, a geodesic between endpoints of the half-circle, and the exponential map with initial velocity tangential to the central half-circle. Bottom row: comparison with LAND on the same set of experiments.

exponential map takes approximately six seconds. For all computations, the initial penalty parameter is set to $\mu_0 = 100$ and the maximum penalty parameter to $\mu_{\max} = 1000$. For the LAND model, we use a locality radius of $\sigma = 0.04$ and a density regularization of $\rho = 10^{-4}$. Computation of all five geodesics with the LAND model takes approximately two minutes, while the exponential map requires about one second. Our parameters for the learned image manifold are as reported in Appendix B.3. The computation of a geodesic takes under two minutes for our model and three minutes for the LAND model.

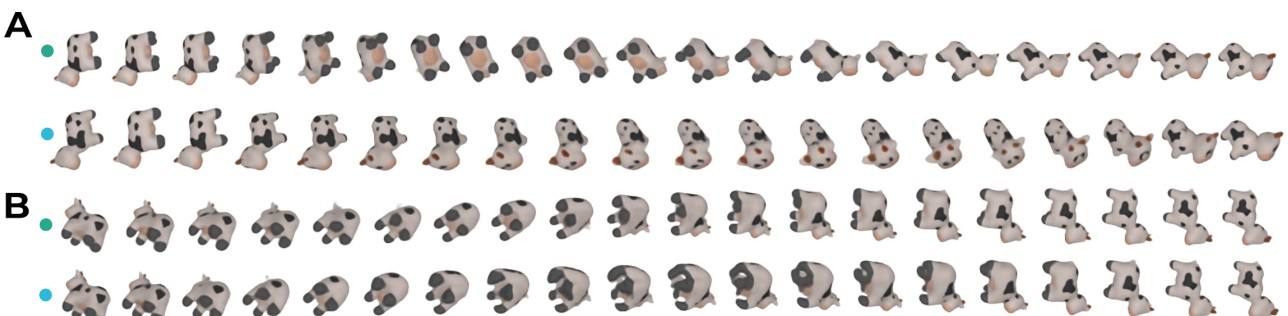

*Figure 18.* Comparison on the latent manifold from Section 5.3. Top rows (green): geodesics computed with our approach from Figure 8. Bottom rows (blue): geodesics computed with the LAND model.

## D. Intuitive introduction to the geometric setting

In this section we provide a more intuitive introduction to the setting of Section 2 and motivate the discretization in Section 3 in the Euclidean case. In particular, we restrict this exposition to manifolds embedded in an higher-dimensional space, which inherit the Euclidean ambient metric. The general setting allowing different metrics is discussed in Sections 2 and 3 and Appendix A.1.

An autoencoder embeds input data into a lower-dimensional latent space. However, the encoded data is usually clustered around an even lower-dimensional latent manifold $\mathcal{Z}$. We represent this manifold implicitly as the zero level set of a function $\zeta$, i.e. $\mathcal{Z} = \{z \in \mathbb{R}^l \mid \zeta(z) = 0\}$. For example, the torus with outer and inner radius $R, r > 0$ is described by $\mathcal{Z} = \{z = (z^1, z^2, z^3) \in \mathbb{R}^3 \mid 0 = \zeta(z) := (\sqrt{(z^1)^2 + (z^2)^2} - R)^2 + (z^3)^2 - r^2\}$. Implicit representations are preferable to explicit parametrizations, which typically only exist locally and are harder to learn from point cloud data.

**Path energy and geodesics.** To interpolate between two encoded data points, a straight line between them might leave the manifold and go through empty space (Figure 2). Instead, we want to find an interpolation path that follows the manifold. The shortest such paths are called geodesics. In the Euclidean case, geodesics can be computed by minimizing the path

energy

$$\mathcal{E}(\mathbf{z}) = \int_0^1 \|\dot{\mathbf{z}}(t)\|^2 \, dt \,,$$

over all paths $\mathbf{z}(t)$ satisfying the constraint $\zeta(\mathbf{z}(t)) = 0$. To enforce the constraints we introduce the Lagrangian $\mathbf{L}(\mathbf{z}, \lambda) = \int_0^1 \|\dot{\mathbf{z}}(t)\|^2 + \lambda(t)\zeta(\mathbf{z}(t)) \, dt$ with Lagrange multipliers $\lambda(t)$. The resulting Euler–Lagrange equation leads to

$$\ddot{\mathbf{z}}(t) = \tfrac{1}{2}\lambda(t)D\zeta(\mathbf{z}(t)) \,, \tag{17}$$

which has a natural geometric interpretation: the acceleration $\ddot{\mathbf{z}}$ is perpendicular to the manifold, $\ddot{\mathbf{z}}(t) \perp T_{\mathbf{z}(t)}\mathcal{Z}$, since $\ker D\zeta(z) = T_z\mathcal{Z}$. It means that the path only changes its direction because the manifold is curved, never accelerating along it. Equation (17) leads to a second-order ODE that, given a start point $\mathbf{z}(0) = z$ and an initial tangential velocity $\dot{\mathbf{z}}(0) = v$, can be solved forward in time. Following this solution for one unit of time leads to the exponential map $\exp_z(v)$ that corresponds to extrapolation in direction $v$.

**Discretization.** To compute geodesics numerically in this setting, we compute the path energy of a discrete path $\mathbf{z} = (z_0, \ldots z_K)$ by

$$\mathcal{E}^K(\mathbf{z}) = K \sum_{k=1,\ldots,K} |z_{k-1} - z_k|^2 \,.$$

Minimizing the energy over interior points $k = 1, \ldots, K-1$ under the constraints $\zeta(z_k) = 0$ leads to the optimality conditions (7)–(8), where (7) simplifies in this setting to

$$0 = K(z_{k+1} - 2z_k + z_{k-1}) - \sum_{i=1,\ldots,l} \Lambda_{ik} \nabla \zeta_i(z_k) \quad \text{for} \quad k = 1, \ldots, K-1, \tag{18}$$

with $\Lambda \in \mathbb{R}^{l,K-1}$ a matrix of Lagrange multipliers. This is the discrete analogue of (17), where in the first term finite differences replace the second derivative. A discrete exponential map at point $z$ and in direction $v$ is then computed by solving these conditions iteratively given two successive points $z_{k-1}, z_k$, starting with $z_0 = z$ and $z_1 = z_0 + \frac{1}{K}v$.

