# OpenReview forum: "Geodesic Calculus on Implicitly Defined Latent Manifolds"
_ICML.cc/2026/Conference — ICML 2026 regular_

### Official Review · Reviewer_iPaz · 2026-02-23

**Soundness:** 3
**Presentation:** 2
**Significance:** 3
**Originality:** 3
**Overall Recommendation:** 3
**Confidence:** 3

**Summary:**

This paper develops a discrete Riemannian calculus approximating classical geometric operators using an augmented Lagrangian. To obtain a suitable implicit representation, this paper proposes to learn an approximate projection onto the latent manifold by minimizing a denoising objective, which is independent of the underlying autoencoder and supports different Riemannian geometries.

**Compliance With Llm Reviewing Policy:**

Affirmed.

**Final Justification:**

The rebuttal solved some of my concerns, but I still think using the projection as a regularization term for geodesic computing may introduce some new problems and cannot solve the fundamental problem of geodesic learning on high-dimensional complex data. Because if I understand correctly, the denoising autoencoder objective yields $\zeta (z)=\nabla\log p_\sigma(z)$ at the optimum, the regularization term effectively enforces $\nabla \log p_\sigma(z)=0$, This intuition is somewhat reasonable, but the argument is not yet rigorous. Moreover, for high-dimensional data, this type of reconstructed-manifold method may introduce substantial challenges.
Thus, I slightly raise my score to 3. But I'm not an expert in this area, I can't judge where the acceptance bar should be, so I'm not very confident of my score.

**Key Questions For Authors:**

1. Comparisons with other related methods should be included.
2. It's better to verify the scalability of the proposed method in high-dimensional datasets.
3. The motivation and rationality for the projection of implicit representation should be clarified.
4. Some notations should be clarified and made consistent. For example, the $\mathcal{Z}$ in the left column of line 160 is different from that in the right column of line 113. Similar inconsistencies in notation can be found below.

**Limitations:**

yes

**Strengths And Weaknesses:**

Strength:
1. An efficient and accurate geodesic calculus is essential in understanding the geometry of the data and latent manifold, which has huge potential for various areas such as generative models.
2. The proposed method is easy to implement and has been shown to be effective in various low and mid-dimensional datasets.

Weakness:
1. The motivation and theoretical justification for the proposed projection of implicit representation are not clarified. Why use a denoising objective to learn the projection? The denoising objective is often used to estimate the score or energy function of the data distribution, which is not directly related to the implicit representation of the latent manifold. And where does the noise come from? Does it originate from the input data itself, or is it artificially added during training? The necessity of the projection should be clarified. Why your projection function can reconstruct the latent manifold and can be used as a regularizer to help avoid the 'holes' in the manifold.
2. The experiments only focus on low or mid-dimensional datasets? How about the scalability on high-dimensional datasets?
3. The experiments only compare the proposed geodesic interpolation with linear interpolation, lacking comparisons with other related methods[1,2].

[1] Arvanitidis, G., Hansen, L. K., and Hauberg, S. Latent space oddity: on the curvature of deep generative models. In ICLR, 2018.
[2] Shao, H., Kumar, A., and Thomas Fletcher, P. The Riemannian geometry of deep generative models. In CVPR Workshops, 2018.

---

> ### Author Rebuttal · Authors · 2026-03-30
>
> Thank you for your feedback. We address your questions in the following.
>
> **Q1.** Due to space constraints we had deferred our comparisons with other methods to the appendix: Sections C1-C4 provide extensive comparisons to existing methods and we refer to these comparisons in Section 1.
> We understand that reviewers are not obliged to read the appendix, but hopefully those comparisons address your concern.
> In particular, we compare with cited method [1] in C3 (see Fig.16 for results).
> As outlined there, this method is limited to VAEs and requires a modification or regularization of their variance as well as a graph structure on the latent manifold to compute geodesics.
> Method [2] relies on a pure pullback metric to define geometry in latent space, which does not enforce staying on the latent manifold.
> As a consequence, it can only reliably compute geodesics if the dimension of latent manifold and latent space coincide,
> which is why in Section C2 we mention the underlying technique of [2], but perform our comparisons with methods that additionally respect the latent manifold.
> Results of pure pullback metrics are shown in Fig.13 (blue paths) and Fig.16 (black crosses).
>
> **Q2.** In our work, we studied latent dimensions up to l=84 (see, e.g., Fig.9), which covers a wide range of use cases in scientific shape analysis and representation learning.
>
> A general scalability analysis has to distinguish two aspects:
>
>   - Optimization via the Augmented Lagrange method. Its per iteration cost depends on the dimensionality of the unconstrained subproblem, but those methods are explicitly promoted for large scale problems [5] and even apply to infinite-dimensional problems [6].
>   - The computation of the implicit representation. It is based on minimizing the denoising loss and thus is dimension-dependent as general learning-based approaches.
> For diffusion models in very high-dimensional settings (whose score typically derives from denoising autoencoders), such problems appear well-behaved under the right design choices.
>
> One advantage of our approach is that, unlike other methods ([1] or also [3],[4]), we do not rely on latent space grids to compute geodesics, which suffer from the curse of dimensionality.
> To summarize, we agree that our current method is designed for latent spaces of moderate dimension.
> However, in contrast to competing methods we do not see a clear bottleneck when scaling up.
>
> **Q3.** We will add a stronger motivational paragraph addressing the corresponding questions in weakness 1.
> As you mentioned, the denoising objective is often used to estimate the score of a data distribution.
> However, this score is directly related to the projection onto the distribution, as it points towards increasing data density, cf. [7].
> This relation is readily illustrated as follows. Assuming for simplicity that the data is Gaussian-distributed around the manifold M, i.e. with density $p(x)\exp(-\text{dist}^2(x,M)/2\sigma^2)$,
> then the score $\sigma^2 \nabla \log(p(x))$ equals exactly $\Pi(x)-x$ for $\Pi$ the projection onto $M$.
> As visualized in Fig.1,2, the learned projection \Pi indeed points to the manifold.
> Finally, this projection $\Pi$ is a particular implicit representation of the manifold: The manifold is just the zero-level set of the map $\text{id}-\Pi$.
> If the manifold has a hole, then inside this hole $\Pi $ points back to the manifold, and this ensures that computed geodesics avoid the hole.
>
> Regarding the noise, there are two types:
> 1. noise inherent in the input data or arising from imperfect embeddings (as in any trained encoder on real data),
> 2. additional Gaussian noise introduced as part of the definition of the denoising objective (Eq. 13), similar to the diffusion model setting.
>
> In section A3 of the appendix, we evaluate the denoising objective for different noise levels in input data (type 1), cf.  (Fig.11 right),
> and different levels of Gaussian noise (type 2), cf. (Fig.11 left) and Fig.10.
>
> **Q4.** Thank you for pointing this out, you are correct. We will fix the definition of Z in a revised version.
> To clarify: the latent manifold is defined as the encoding  $\phi(M)$ of the unknown manifold M
> and is the zero level set of a representation function $\zeta$ on the latent space.
>
> We hope this answers your questions.
>
> [3] Arvanitidis, G., Gonzalez-Duque, M., Pouplin, A., Kalatzis,D., and Hauberg, S. Pulling back information geometry. AISTATS 2022
>
> [4] Arvanitidis, G., Hansen, L. K., and Hauberg, S. A locally adaptive normal distribution. NEURIPS 2016
>
> [5] Birgin, Ernesto G.,Jose Mario Marti­nez. Practical augmented Lagrangian methods for constrained optimization. SIAM, 2014.
>
> [6] Kanzow, C., Steck, D., Wachsmuth, D. An Augmented Lagrangian Method for Optimization Problems in Banach Spaces. SICON, 2018.
>
> [7] Kharitenko, A., Shen, Z., de Santi, R., He, N., & Doerfler, F. Landing with the Score: Riemannian Optimization through Denoising. arXiv preprint arXiv:2509.23357. accepted at ICLR 2026

---

> > ### Author Rebuttal · Reviewer_iPaz · 2026-04-01
> >
> > Thanks for the authors’ rebuttal and efforts. But some of my concerns have not been resolved. First, for the experimental part, the dimension of the latent manifold seems very important, but the paper didn't discuss this. I guess the latent dimension cannot be too large. And in this work, the latent dimensions are up to 84, which is too small to satisfy nowadays's high-dimensional complex datasets. And no experiments support the scalability of the proposed method.
> >
> > For the theoretical part, we thank the authors for clarifying the motivation of the proposed denoising loss function. But from my understanding, the augmented Lagrangian method seems to add a regularization term to restrict the geodesic within the latent manifold, but this can't solve the problem of avoiding 'holes' in the data manifold, as illustrated in Fig.6, it depends on Riemannian metrics rather than the augmented term. Additionally, if I understand correctly, the regularization term constrains the geodesic to stay as close to the projection $\zeta$ as possible, which may introduce deviations when decoding back to the original data space.
> >
> > So I still can't fully understand the benefit attributed to the learned projection and the augmented Lagrangian method.
> > Could the authors explain these points?

---

> > > ### Author Response · Authors · 2026-04-06
> > >
> > > Thank you for taking the time to critically evaluate our rebuttal.
> > >
> > > Concerning your point relating to the learned projection and the augmented Lagrangian method:
> > > The observation that the augmented Lagrangian method adds a regularization term to restrict the geodesic within the latent manifold is correct. This regularization term is, in turn, based on the learned projection. Hence, if the learned projection avoids holes in the data manifold, so will the computed geodesics. This is independent of the chosen metric. Fig. 6 demonstrates this, as for two different metric (curves in yellow and green) the hole in the middle is avoided through which the linear interpolation (in red, same metric as yellow) passes straight through. The projection is learned in a fashion such that we stay on the latent manifold and thus when decoding should yield sensible data points on the data space with minimal deviations.

---

### Official Review · Reviewer_j8Tj · 2026-03-02

**Soundness:** 3
**Presentation:** 3
**Significance:** 3
**Originality:** 3
**Overall Recommendation:** 4
**Confidence:** 3

**Summary:**

The latent spaces of autoencoders are low dimensional embeddings of the ambient spaces, hence can be studied from a geometric point of view. The paper proposes to model the latent spaces as implicit submanifolds. A denoising objective is employed to learn the projection, which is independent of the underlying autoencoders and the geometric structures of the latent spaces. Algorithms for computing time-discrete geodesic curves are provided. Empirical evidence are provided, highlighting the benefits of the proposed approach.

**Compliance With Llm Reviewing Policy:**

Affirmed.

**Final Justification:**

The authors provided reasonable answers to my concerns, and I maintain my original positive rating.

**Key Questions For Authors:**

1. While I imagine there could be some, I could not find any explicit statements regarding why we should look into implicit manifolds and why they improve upon the existing method. It would be beneficial if the authors can provide further motivation.

2. As an alternative to using implicit manifolds, one could learn explicit coordinates and perform the geometric operations using the coordinates. When both are applicable, would one of them be better than the other?

**Limitations:**

yes

**Strengths And Weaknesses:**

Strengths:

1. I find it an interesting idea to model the latent space as implicit submanifolds.

2. Empirical evidence demonstrates that the proposed approach provides paths that remain meaningful.

Weaknesses:

1. Some parts could be written more clearly to better motivate the work.

2. Some additional ablation studies could be interesting.

Please refer to Key Questions 1 and 2 for further details on Weaknesses 1 and 2, respectively.

---

> ### Author Rebuttal · Authors · 2026-03-30
>
> Thank you for your feedback and comments. We address your two key questions below.
>
> **Q1** We propose to model latent manifolds as implicit manifolds in order to have a unified framework for latent geometry.
> Existing methods, which we compare to in appendix C3 and C4, are typically applicable only in restricted settings.
> Those methods often try to derive a distance function on the full latent space,
> which is less robust than a projection as it in principle also allows interpolation paths to leave the latent manifold.
> We believe that directly constraining computation to the manifold is conceptually cleaner and more robust.
> Moreover, it allows an efficient way to perform computation of geodesics that does not require to calibrate the decoder
> (which would restrict to specific autoencoders)
> or to compute latent grids (which is only feasible in low latent dimensions) as the other methods presented in that section.
> The obvious further alternative to learn a parametrization of the latent manifold is harder to obtain, see the next answer.
>
> We would move some of this discussion from the appendix to the main text to make it more explicit and accesible to readers as we acknowledge that the motivation of this approach in the main text falls a bit short.
>
> **Q2** When a parametric and an implicit representation are available, the computation of geodesics is similar and can be performed with the same framework.
> The parametric viewpoint has the advantage that the computation takes place in a lower-dimensional parameter domain.
> We give an explicit comparison in appendix C1 (Comparison to a parametric viewpoint).
> However, in contrast to implicit representations, global parametrizations of manifolds do not exist in general due to topological obstructions.
> Furthermore, learning a local parametrization of a manifold from point cloud data is a substantially harder problem,
> and we are not aware of practical research that addresses this in general settings.
>
> Concerning the comment on ablation studies we agree on their usefulness.
> Let us point out that example B.4 (Pose-NDF) actually represents an ablation study replacing our learned projection with a neural distance function
> and that in C1 we replace the geodesic computation on an implicit manifold representation by one on a parametrized manifold.
> Further ablation studies would include replacing our denoising-based projection by more advanced score-based or diffusion models, an  interesting direction for future work.
>
> We hope this answers your questions. We are happy to include an improved motivation of our approach with respect to
> the above mentioned key points in the main text.

---

> > ### Author Rebuttal · Reviewer_j8Tj · 2026-04-03
> >
> > I thank the authors for addressing my questions. As a comment on "learning a local parameterization of a manifold from point cloud data", there are works on that direction; see the following references, to name a few. While I agree the implicit representations provide an alternative to such methods, the authors might consider to discuss works along this direction, and further discuss the pros and cons of using implicit representations.
> >
> > [1] Chart Auto-Encoders for Manifold Structured Data, Schonscheck et al.
> >
> > [2] Density estimation on smooth manifolds with normalizing flows, Kalatzis et al.
> >
> > [3] VQ-Flows: Vector Quantized Local Normalizing Flows, Sidheekh et al.
> >
> > [4] Manifold Learning by Mixture Models of VAEs for Inverse Problems, Alberti et al.
> >
> > [5] Learning Geometry and Topology via Multi-Chart Flows, Yu et al.

---

> > > ### Author Response · Authors · 2026-04-07
> > >
> > > Thank you for considering our rebuttal and for pointing us to these works.
> > >
> > > We will expand the discussion in the appendix to further clarify the respective limitations and advantages of learned parametric versus learned implicit representations.

---

### Official Review · Reviewer_H4DV · 2026-03-07

**Soundness:** 3
**Presentation:** 2
**Significance:** 3
**Originality:** 3
**Overall Recommendation:** 4
**Confidence:** 2

**Summary:**

This paper proposes a way to do geodesic calculus on implicit latent manifolds learned by autoencoders so that latent interpolation will be on the geodesic. The key idea is to learn an approximate projection from latent space onto the latent manifold using a denoising objective (like in diffusion models), then use that implicit representation inside a time-discrete variational geodesic framework to compute interpolating geodesics and discrete exponential maps. The paper argues that this avoids committing to a decoder-specific geometric formulation and allows different Riemannian metrics to be used on the latent manifold. It evaluates the approach on synthetic and real examples, including motion capture with spherical VAE structure, and image data with low-distortion autoencoders.

**Compliance With Llm Reviewing Policy:**

Affirmed.

**Final Justification:**

The method is mathematically well-motivated and robust to imperfect representations, and it has clear potential impact in structured latent spaces. However, limitations remain in terms of clarity (the paper is hard to follow for non-experts) and scalability to high-dimensional latent spaces. The rebuttal constructively addressed my main concerns, particularly regarding sparsity, modularity, and partial scalability, while acknowledging remaining limitations and committing to improved exposition. Overall, the rebuttal reinforced my initial assessment: this is a solid and interesting contribution with some scope limitations, and I therefore maintain a Weak Accept recommendation.

**Key Questions For Authors:**

- How large can latent dimension get before the optimizer becomes impractical?
- Can the learned denoising projection be shared across tasks or must it be retrained per autoencoder/data manifold?
- How sensitive is the method to sparse sampling of the encoded point cloud?
- Could score models / diffusion denoisers replace the projection network more effectively, as hinted in the conclusion?

**Limitations:**

Not relevant

**Strengths And Weaknesses:**

*Strengths*:
- The paper identifies a real gap in the latent-geometry literature: many prior methods either rely on explicit decoder parameterizations or make it hard to do practical interpolation/extrapolation that both stays near the data support and respects task-relevant geometry.
- Using a time-discrete path-energy viewpoint is mathematically elegant. It avoids directly integrating geodesic ODEs .
- The paper does not assume a perfect implicit representation. In fact, much of the numerical method is justified around the observation that the learned constraint is inexact.

*Weaknesses*:
- The paper is hard to understand to a non-expert in Differential Geometry and topology.
- The method involves: learning a separate projection network and solving constrained optimization for each geodesic using augmented Lagrangian loops. That is probably fine for low-dimensional latent manifolds and scientific shape spaces, but the paper does not yet make a compelling case for large latent spaces typical of modern representation learning or generative foundation models.
- The denoising projection (as in diffusion models) requires a strong prior and therefore many training examples. It is unclear how well this method will perform when the number of samples is sparse.

---

> ### Author Rebuttal · Authors · 2026-03-30
>
> Thank you for your detailed feedback. We will address your four questions and along the way also respond to your mentioned weaknesses.
>
> **Q1** In our work, we studied latent dimensions up to $l=84$ (see, e.g., Fig.9), which covers a wide range of use cases in scientific shape analysis and representation learning.
> A scalability analysis beyond these experiments would have to distinguish two aspects:
>
>   (i) Optimization via the Augmented Lagrange method. Its per iteration cost of course depends on the dimensionality of the unconstrained subproblem, but those methods are explicitly promoted for large scale problems [4] and even apply to infinite-dimensional problems [5].
>
>   (ii) The computation of the implicit representation. It is based on minimizing the denoising loss and thus is dimension-dependent as general learning-based approaches. For diffusion models in very high-dimensional settings (whose score typically derives from denoising autoencoders), such problems appear well-behaved under the right design choices.
>
> One advantage of our approach is that unlike other methods ([1],[2],[3]), which we compare to in appendix C2-C4, we do not rely on latent space grids to compute geodesics, which suffer from the curse of dimensionality.
>
> **Q2** The implicit representation has to be learned for each example as it is a specific representation for one specific latent manifold.
> However, standard training procedures apply, and retraining is straightforward for different autoencoders or different data.
>
> **Q3** The method is not strongly sensitive to sparse sampling. We considered examples with varying sampling density (cf. Fig.6).
> From the mathematical viewpoint, our method is designed to work with different degrees of sparsity by tuning the denoising parameter $\sigma$ (larger $\sigma$ for sparser data).
> In principle, it allows to obtain latent manifold approximation adapted to the degree of sparsity, i.e. to extract as much information as possible from the sparse data.
> A more refined approach would be to adapt this parameter locally based on local density estimates,
> which would be an interesting direction to study.
>
> **Q4** In our considered examples the denoising objective performed well - it has the great advantage of being relatively easy to train.
> However, for future work it would be indeed interesting to explore more advanced score-based models or diffusion models.
> Since our framework and code base are modular, it is straightforward to integrate alternative representation functions.
>
> Regarding the weakness on accessibility:
> We acknowledge that for readers from different communities the differential geometry setting may be challenging.
> We are happy to add an extended introductory section to the appendix that explains the key notions of differential geometry and their concrete meaning in this context at a more accessible level.
>
> We hope this addresses your questions.
>
> [1] Arvanitidis, G., Hansen, L. K., and Hauberg, S. Latent space oddity: on the curvature of deep generative models.
>
> [2] Arvanitidis, G., Gonzalez-Duque, M., Pouplin, A., Kalatzis,D., and Hauberg, S. Pulling back information geometry. AISTATS 2022
>
> [3] Arvanitidis, G., Hansen, L. K., and Hauberg, S. A locally adaptive normal distribution. NEURIPS 2016
>
> [4] Birgin, Ernesto G.,Jose Mario Marti­nez. Practical augmented Lagrangian methods for constrained optimization. Society for Industrial and Applied Mathematics, 2014.
>
> [5] Kanzow, C., Steck, D., Wachsmuth, D. An Augmented Lagrangian Method for Optimization Problems in Banach Spaces. SICON, 2018.

---

> > ### Author Rebuttal · Reviewer_H4DV · 2026-04-04
> >
> > Thank you for the clear and detailed responses.
> >
> > Q1 (Scalability):
> > The clarification is helpful. The method appears suitable for moderate latent dimensions. However, the practical limits in higher-dimensional latent spaces remain somewhat unclear and could be better scoped in the paper.
> >
> > Q2 (Transferability):
> > It is now clear that the implicit representation must be learned per manifold. This is reasonable, though it limits transferability across models.
> >
> > Q3 (Sparse Sampling):
> > The explanation regarding tuning the denoising parameter is convincing, and the reference to existing experiments is helpful. Still, robustness to sparsity could be emphasized more explicitly in the paper.
> >
> > Q4 (Alternative Models):
> > The modularity of the framework is a strength, and the possibility of integrating score/diffusion models is well justified without overclaiming.
> >
> > I appreciate the authors’ willingness to improve the exposition. Adding an accessible introduction to the geometric concepts would significantly strengthen the paper.
> >
> > Overall:
> > The rebuttal addresses my main concerns. Some questions remain regarding scalability and clarity of scope, but I maintain my original assessment: the work is technically sound, original, and a meaningful contribution. I remain supportive of acceptance.

---

### Official Review · Reviewer_ctXU · 2026-03-09

**Soundness:** 2
**Presentation:** 2
**Significance:** 3
**Originality:** 3
**Overall Recommendation:** 4
**Confidence:** 2

**Summary:**

The authors propose a practical method for computing geodesic interpolation in the latent spaces of various autoencoders.
In previous studies, the computation of geodesic interpolation was possible only for specific autoencoders. In contrast, this study makes it possible to compute geodesic interpolation in the latent spaces of various autoencoders by learning an approximate projection onto the latent manifold using a denoising objective.
As a result, the authors claim that their method achieves more realistic interpolation than linear interpolation.

**Compliance With Llm Reviewing Policy:**

Affirmed.

**Final Justification:**

Since the key questions have been largely resolved through the discussion with the authors, I would like to revise my score.

**Key Questions For Authors:**

Does the claim in the introduction, stated as a contribution, that geodesic interpolation and extrapolation have been made possible mean only that they are achievable to the extent demonstrated at the level of visualization? Or does the paper intend to conclude, from the visualization-based evidence, that geodesic interpolation and extrapolation are generally possible?

The conclusion states that efficient computation requires fast distance evaluation in the data manifold or an embedding of $Z$ that is (near-)isometric. Under what circumstances can fast distance evaluation in the data manifold be assumed?

**Limitations:**

yes

**Strengths And Weaknesses:**

Strengths:
* Soundness: The feasibility of geodesic interpolation / extrapolation is demonstrated on different types of examples, namely discrete shells, motion capture, and images.
* Presentation: The experimental results are visually easy to understand, and it is clearly shown that the proposed method improves over linear interpolation.
* Significance: By learning an implicit representation using a denoising objective, the paper proposes a method applicable to various autoencoders. It also obtains realistic results on a variety of examples in practice.
* Originality: The paper enables discrete geodesic calculus in an implicit representation by learning a denoising objective.

Weaknesses:
* Soundness: In SCAPE, the paper shows avoidance of self-intersection; in motion capture, realistic decoded paths; and in Pose-NDF, consistency with a plausible pose manifold. However, all of these evaluations are primarily visualization-based. If the ultimate goal is limited to these objectives, the paper may be considered sufficiently sound. However, it remains unclear whether these results alone are enough to support the contribution claimed in the paper, namely, ``a time-discrete geodesic calculus for imperfect representations of implicit latent manifolds and provide algorithms for geodesic interpolation and extrapolation.'' The paper should clarify its ultimate goal and more clearly delimit the scope of what its results enable.
* Presentation: Although the conclusion states that geometric operations on a latent manifold are feasible, the experiments mainly provide visualization-based evidence. It is therefore unclear whether the paper uses “feasible” only in this limited sense or intends to claim a broader conclusion.
* Significance: The conclusion states that efficient computation requires fast distance evaluation in the data manifold or an embedding of Z that is (near-)isometric. It would be desirable to clarify to what extent these requirements become bottlenecks in practical applications.
* Originality: There is already substantial prior work on the general theme of handling geometry in latent spaces, as also reflected in the related-work section. Thus, the originality lies less in the overall theme and more in the method itself. Still, the method is novel in being applicable to various autoencoders, and this is not a major weakness as long as its limitations are clearly stated.

---

> ### Author Rebuttal · Authors · 2026-03-30
>
> Thank you for your detailed feedback on our work. We address your two questions below.
>
> **Q1** This is a valid point, and we will state our scope more clearly in a revised version.
> Indeed, in our results section, the evaluation of geodesics is visualization-based.
> This is because no ground truth is available.
> Even worse, in some of our application examples, there may not even exist a perfectly well-defined data manifold, which would be prerequisite for the computation of geodesics.
> Yet, our method even yields reasonable results in those cases (from the application and visual inspection viewpoint) superior to other existing methods as the comparisons in Sections C3, C4 show.
> Therefore, a more careful description of our scope could be:
> "We provide a framework to compute realistic inter- and extrapolation on latent manifolds.
> For regular, smooth data manifolds this framework corresponds to geodesic inter- and extrapolation with respect to a Riemannian metric for which there are multiple, flexible choices (see appendix A1)."
>
> In the setting of regular smooth data manifolds one may ask whether our numerical approach converges to true geodesics, and in the revision we will also comment on that more clearly.
> Our method consists of two building blocks:
>  1. learning the implicit representation,
>  2. performing the geodesic computation.
>
> Regarding (1.) it is shown in [1] that, in an idealized setting, the minimizer of the denoising objective has an error of order $O(\sigma^2)$. A full mathematical error analysis is beyond the scope of our paper, but instead Fig.11 provides empirical error evaluations for different parameters.
>
> Concerning (2.), assuming a ground truth implicit representation, [2] gives convergence results and error estimates for computed geodesics and the exponential maps.
>
> To rigorously show that our method makes geodesic computations on latent manifolds possible, one would have to control the error propagation combining both steps (1.) and (2.).
> As a consequence, the total error of the geodesic computation could be bounded in terms of the input data and sampling noise, the denoising parameter $\sigma$, and the discretization parameter $K$.
> We expect this to hold for $\sigma$ sufficiently large compared to $1/K$, but without an explicit error analysis in this paper.
> Instead, in Fig.3 & 4, we show for particular examples, where a ground truth is available, that without input data noise our method converges to ground truth results.
>
> **Q2** This addresses a difficulty that any method for latent geometry has to deal with.
> Our approach is flexible in its choice of the metric and we only require a local distance evaluation.
> Both features increase the chances in applications to find an appropriate distance with fast evaluation.
> In fact, in many applications such as our shape space examples there are feasible distances or local distance approximations.
> In cases where such an evaluation is infeasible, one could approximate the distance by a learned function, which might be a promising direction for future work.
>
> We hope this answers your questions.
>
> [1] Alain, G., & Bengio, Y. What regularized auto-encoders learn from the data-generating distribution. The Journal of Machine Learning Research (2014)
>
> [2] Rumpf, M. and Wirth, B. Variational time discretization of geodesic calculus. IMA Journal of Numerical Analysis (2012)

---

> > ### Author Rebuttal · Reviewer_ctXU · 2026-04-02
> >
> > Thank you for your comments.
> >
> > A1: We understand your point as asking whether our claim is limited to what can be confirmed at the level of visualization. We believe this issue is resolved by making the statement more explicit in the revised manuscript.
> >
> > A2: Thank you for your response. Our question may not have been communicated clearly. In the conclusion of the paper, there is the statement: " Achieving efficiency, however, requires either fast distance evaluation in the data manifold or an embedding of Z that is (near-)isometric." This seems to imply that the proposed method in this paper cannot be implemented efficiently unless there exists a nearly isometric embedding. What I wanted to ask was: for what class of problems can one assume the existence of a nearly isometric embedding? In other words, I would like to know whether this assumption can be made for many problems and therefore is not much of a bottleneck, or whether it is in fact a strong assumption.

---

> > > ### Author Response · Authors · 2026-04-06
> > >
> > > Thank you for reading our rebuttal and providing further feedback.
> > >
> > > Concerning A2:
> > > Answering directly to your question about the existence of nearly isometric embeddings: This is covered by the Nash-Kuiper theorem, which (very briefly) states that such embeddings always exist. Finding/learning them is then, of course, still a challenging problem. However, various methods have made progress on this for data manifolds.
> > > Alternatively, one could work with non-isometric embeddings, as we also propose, as long as the local distance is cheap to evaluate. For many shape spaces that motivates our work, this is indeed possible. Combined, we believe that our method can be applied to many interesting (data) manifolds.

---

### Decision · Program_Chairs · 2026-04-30

**Decision:**

Accept (regular)

**Comment:**

The reviewers agree that the paper brings forward an interesting methodology for an understudied problem. They raise a series of concerns that would be good to address. In particular, a more open discussion about the importance of the latent dimension seems to be missing. I encourage the authors to take the provided feedback into account when revising the paper.